# TokenDrop: Token-Level Importance-Aware Backward Propagation Skipping for Efficient LLM Fine-Tuning

**Beomseok Kim** [1]   **Sol Namkung** [1]   **Dongsuk Jeon** [1]

## Abstract

Despite the success of parameter-efficient fine-tuning (PEFT) methods in reducing parameter-related overhead, fine-tuning large language models (LLMs) is still bottlenecked by significant memory and computational demands. In this paper, we propose **TokenDrop**, a token-level importance-aware backpropagation skipping method that reduces activation memory and accelerates LLM fine-tuning by skipping backward computations for less informative tokens. TokenDrop evaluates token importance based on the magnitude of residual updates during the forward pass, enabling lightweight, gradient-free importance estimation. Furthermore, we introduce cumulative token selection to preserve gradient continuity across layers and lazy selection scheduling that defers token selection to facilitate globally informed importance scoring under memory constraints. Across a range of experiments, TokenDrop achieves up to **42.9**% reduction in memory usage and up to **1.50**× training speedup, while preserving accuracy and outperforming existing backpropagation-skipping baselines. The code is available at https://github.com/kimbss470/tokendrop_official.

## 1. Introduction

Large language models (LLMs) have demonstrated exceptional capabilities across a broad spectrum of natural language tasks (Achiam et al., 2023; Team et al., 2023; Touvron et al., 2023; Guo et al., 2025). To adapt these general-purpose models to specific domains, fine-tuning is widely employed to obtain task-specific knowledge and instruction-following abilities. However, as the scale of LLMs and

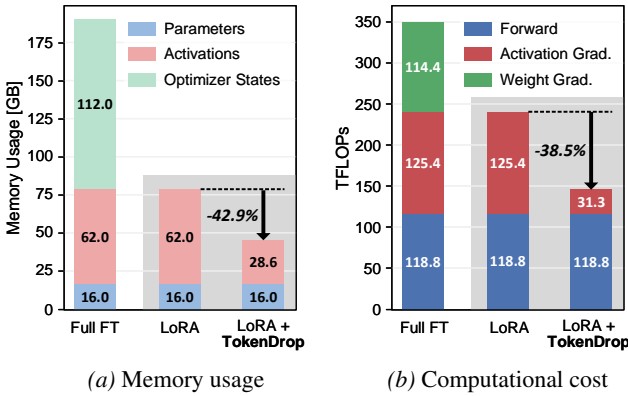

*(a)* Memory usage          *(b)* Computational cost

*Figure 1.* Breakdown of (a) memory usage and (b) computational cost of conventional and proposed fine-tuning algorithms when fine-tuning LLaMA3.1-8B with a micro-batch size of 16 and a maximum sequence length of 1024. TokenDrop significantly reduces the memory and computational costs of LoRA.

their context window sizes continue to grow, the fine-tuning process becomes increasingly resource-intensive, imposing significant burdens on memory footprint and computational budgets.

To mitigate these challenges, Parameter-Efficient Fine-Tuning (PEFT) methods, such as LoRA (Hu et al., 2022) and QLoRA (Dettmers et al., 2023), have been widely adopted. By freezing pretrained weights and updating only a small subset of trainable parameters, PEFT significantly reduces the memory required for parameter states and effectively lowers the FLOPs associated with weight gradient computation. However, as shown in Figure 1, despite these advancements in parameter-related efficiency, a critical bottleneck remains: activation-related costs. Since PEFT still requires a full forward pass and backpropagation, it does not alleviate the activation memory burden, and the computational overhead of calculating activation gradients remains substantial.

To address these remaining bottlenecks, recent work has explored skipping the backward pass during fine-tuning, as illustrated in Figure 2. DropBP (Woo et al., 2024b) accelerates fine-tuning by stochastically skipping backward propagation at the block level. Here, we refer to the multi-head attention (MHA) or feedforward network (FFN) module

---

[1]Department of Intelligence and Information, Seoul National University, Seoul, South Korea. Correspondence to: Dongsuk Jeon <djeon1@snu.ac.kr>.

*Proceedings of the 43rd International Conference on Machine Learning*, Seoul, South Korea. PMLR 306, 2026. Copyright 2026 by the author(s).

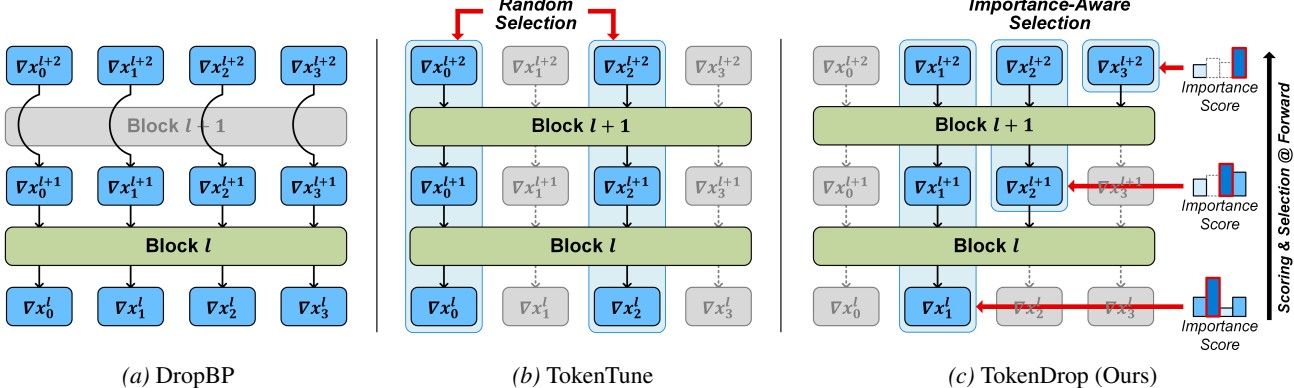

*Figure 2.* Comparison of backpropagation skipping methods. (a) DropBP skips backpropagation at the block level without token discrimination, discarding all token gradients in stochastically dropped blocks. (b) TokenTune randomly selects a fixed subset of tokens to retain and skips backpropagation for unselected tokens across all blocks, without considering token importance. (c) In contrast, TokenDrop selects tokens based on importance scores and enforces that the retained token set persists across subsequent blocks. Importance scores are computed during the forward pass, enabling activation eviction for redundant tokens and thereby reducing memory usage.

within a Transformer layer as a *block*. By bypassing the gradient calculation for dropped blocks, DropBP avoids storing the corresponding activations, thereby reducing both activation memory and training time. On the other hand, TokenTune (Simoulin et al., 2024) performs backpropagation skipping at the token level via random selection. While both approaches yield efficiency gains, coarse-grained gradient skipping (Woo et al., 2024b) or random selection of tokens without considering importance (Simoulin et al., 2024) fail to account for the heterogeneous contribution of individual tokens during fine-tuning, leading to suboptimal performance.

To overcome these limitations, we present TokenDrop, a token-level importance-aware fine-tuning method for memory- and time-efficient LLM fine-tuning. TokenDrop reduces training cost by selectively propagating gradients only through informative tokens while skipping backpropagation for redundant ones. To enable lightweight yet effective token importance identification, we introduce a residual $\ell_2$-norm-based importance criterion, motivated by the residual architecture of LLMs. Building upon this metric, we further enhance training stability and the quality of token selection via two key strategies: (i) **cumulative token selection**, which ensures that any token retained at an earlier block is always retained in all subsequent blocks to preserve gradient integrity and (ii) **lazy selection scheduling**, which enables globally informed selection by deferring selection decisions.

We conduct extensive experiments across multitask language understanding, commonsense reasoning, instruction following, mathematical reasoning, and code generation benchmarks to evaluate TokenDrop. Across diverse fine-tuning scenarios, TokenDrop simultaneously decreases both activation memory footprint and backward computation,

achieving up to 42.9% total memory reduction and up to 1.50× wall-clock speedup, while maintaining accuracy.

## 2. Related Work

**Token Redundancy.** Recent studies have demonstrated substantial redundancy at the token level in LLMs during both training and inference, highlighting the need for selective token processing. RHO-1 (Lin et al., 2024) reveals that only a small fraction of tokens consistently contribute to loss reduction during pretraining, and achieves higher accuracy across various tasks by masking the loss of low-impact tokens. Similarly, Token Cleaning (Pang et al., 2025) extends this line of work to fine-tuning, filtering uninformative or harmful tokens to improve the model performance. While being effective for data quality control, both approaches require auxiliary reference model training and operate by modifying the token-level supervision rather than the backward process itself. Consequently, they do not reduce activation memory or training time.

A parallel line of research focuses on inference acceleration, leveraging token redundancy. Earlier methods (Goyal et al., 2020; Wang et al., 2021; Kim et al., 2022) primarily rely on attention scores to identify and prune less important tokens. However, these methods are largely validated on non-generative NLP tasks and face integration challenges with modern fused attention kernels such as FlashAttention (Dao et al., 2022; Dao, 2024). On the other hand, recent approaches (Yan et al., 2025; Tao et al., 2025) leverage gradient-based saliency, estimated through small predictors trained on auxiliary datasets since gradients are unavailable during inference. However, directly applying these methods to fine-tuning is cumbersome, as they require an additional training phase and, most importantly, irreversible

information loss induced during the forward pass can result in significant performance degradation when applied to fine-tuning (Woo et al., 2024b). In contrast, TokenDrop shares the intuition of exploiting token importance but fundamentally differs in objective and mechanism. TokenDrop preserves the full forward computation while selectively skipping the backward pass at the token level, reducing activation memory and backward computations during fine-tuning.

**Efficient Sparse Training.** Sparse training techniques aim to reduce computational or memory overhead by exploiting sparsity in weights, activations, or attention patterns. Prior studies (Hubara et al., 2021; Zhang et al., 2023; Mozaffari et al., 2024) have introduced N:M sparsity into weights to leverage specialized hardware in modern GPU architectures, effectively reducing training time. On the other hand, SparseLoRA (Khaki et al., 2025) identifies inactive channels using a training-free SVD-based sparsity estimator and sparsifies the corresponding pretrained weight channels to reduce fine-tuning latency, while keeping LoRA parameters intact. Beyond weight-level optimization, LongLoRA (Chen et al., 2024) focuses on the attention mechanism, employing shifted sparse attention to mitigate the quadratic complexity of self-attention for context length extension.

Despite these advancements, weight-centric approaches do not reduce the storage for activations (Hubara et al., 2021; Zhang et al., 2023; Mozaffari et al., 2024; Khaki et al., 2025), and LongLoRA is primarily limited to extreme long-context scenarios where the self-attention mechanism itself is the dominant bottleneck. In contrast, TokenDrop reduces both activation memory and training time, regardless of the sequence length.

**Backpropagation Skipping Method.** The concept of skipping the backward pass has emerged as a promising direction to alleviate activation memory and FLOPs during LLM fine-tuning. DropBP (Woo et al., 2024b) and Token-Tune (Simoulin et al., 2024) maintain a full forward pass while selectively bypassing the backward pass to reduce activation storage and backward computations. Specifically, DropBP introduces a block-level backpropagation skipping strategy, in which the backward pass of individual blocks is stochastically dropped according to drop rates determined via sensitivity analysis. By skipping the gradient computation of certain blocks selected before iteration, it can also avoid storing activations required for backward computation in the dropped blocks. On the other hand, TokenTune employs token-level backpropagation skip, relying on random token selection. Despite their efficiency gains, both methods overlook heterogeneous token importance during fine-tuning, and this results in suboptimal task performance.

Unlike prior work, TokenDrop introduces importance-aware token-level backward skipping, replacing block-level or random strategies with principled token selection that preserves downstream performance, while achieving memory saving and wall-clock speedup.

## 3. Methodology

In this section, we detail the proposed TokenDrop method. We first revisit the backpropagation algorithm to identify opportunities for reducing activation memory and backward computation through token-level backward skipping. We then introduce the core components of TokenDrop: a residual $\ell_2$-norm-based token importance score, cumulative token selection for stable gradient propagation, and lazy selection scheduling for globally informed retention under activation memory budgets.

### 3.1. Efficient Backpropagation via TokenDrop

To illustrate the costs of backpropagation in terms of memory usage and computations, we consider a linear layer, which constitutes the dominant component of the training time in LLM fine-tuning (Khaki et al., 2025):

**Forward Pass:**

$$\mathbf{Y} = \mathbf{X}\mathbf{W}^{\top}, \tag{1}$$

**Backward Pass:**

$$\mathbf{G}_W = \mathbf{G}_Y^{\top}\mathbf{X}, \tag{2}$$

$$\mathbf{G}_X = \mathbf{G}_Y\mathbf{W} \tag{3}$$

Here, $\mathbf{X} \in \mathbb{R}^{T \times h_{\text{in}}}$, $\mathbf{Y} \in \mathbb{R}^{T \times h_{\text{out}}}$, and $\mathbf{W} \in \mathbb{R}^{h_{\text{out}} \times h_{\text{in}}}$ denote the input activations, output activations, and weights, respectively. $T$ represents the sequence length, while $h_{\text{in}}$ and $h_{\text{out}}$ denote the input and output hidden dimensions. Gradients with respect to each tensor are denoted by $\mathbf{G}$.

A key observation is that computing the weight gradients $\mathbf{G}_W$ in Eq. (2) requires input activations $\mathbf{X}$, which must therefore be stored during the forward pass. In addition, since both activation storage and backward computation scale linearly with the sequence length $T$, selectively skipping the backward pass for less important tokens can reduce activation memory footprint and backward computational cost proportionally to the number of discarded tokens.

To this end, TokenDrop selectively performs backpropagation only for a subset of salient tokens. Let $\mathcal{S} \subset \{1, \dots, T\}$ denote the index set of retained tokens, where $|\mathcal{S}| = \widetilde{T} < T$. We define $\mathbf{X}_{\mathcal{S}} \in \mathbb{R}^{\widetilde{T} \times h_{\text{in}}}$, $\mathbf{G}_{X,\mathcal{S}} \in \mathbb{R}^{\widetilde{T} \times h_{\text{in}}}$, and $\mathbf{G}_{Y,\mathcal{S}} \in \mathbb{R}^{\widetilde{T} \times h_{\text{out}}}$ as the sub-matrices obtained by selecting only the rows indexed by $\mathcal{S}$. Under the TokenDrop framework, the backward pass for a linear layer is reformulated as follows:

$$\mathbf{G}_W = \mathbf{G}_{Y,\mathcal{S}}^\top \mathbf{X}_\mathcal{S}, \quad \mathbf{G}_{X,\mathcal{S}} = \mathbf{G}_{Y,\mathcal{S}} \mathbf{W}. \tag{4}$$

By confining the backward computation to $\mathcal{S}$, the weight gradients become dependent solely on the retained activations $\mathbf{X}_\mathcal{S}$. This eliminates the need to store the activations of discarded tokens during the forward pass. Consequently, both activation memory usage and backward FLOPs are reduced by a factor of $T/\widetilde{T}$.

Importantly, the token index set $\mathcal{S}$ is shared across all operations within a block (i.e., MHA or FFN) under TokenDrop. This uniformity ensures consistent gradient flow and maintains the integrity of the chain rule, as $\mathbf{G}_{X,\mathcal{S}}$ and $\mathbf{G}_{Y,\mathcal{S}}$ remain properly coupled throughout the backward pass within the block. However, for self-attention, even when the backward pass of a token as a query is skipped, gradients with respect to key and value of that dropped token are not strictly zero. This arises because, in the backward computational graph, these key and value gradients remain connected to the gradients with respect to attention outputs of other retained tokens. Empirically, we find that omitting these gradients incurs negligible degradation in accuracy, as detailed in Appendix A.

### 3.2. Lightweight Token Importance Scoring

To maintain model accuracy under selective backpropagation, it is critical to define a robust criterion for determining token importance. In the TokenDrop framework, the score must satisfy two primary requirements:

**i) Computational Lightness:** The scoring procedure must be lightweight to ensure that its overhead does not significantly undermine the gains achieved by skipping backward computations.

**ii) Forward-time Identifiability:** The score must be computable during the forward pass so that activations of redundant tokens can be discarded on the fly before they exceed the target memory budget.

In this regard, commonly used criteria such as attention-score-based (Goyal et al., 2020; Wang et al., 2021; Kim et al., 2022) or gradient-based metrics (Kindermans et al., 2017; Mohebbi et al., 2021; Yan et al., 2025; Tao et al., 2025) face practical limitations. The metrics based on attention scores require explicit access to attention maps, which is incompatible with modern fused kernels such as FlashAttention (Dao et al., 2022; Dao, 2024). Therefore, they forfeit the memory and speed benefits provided by fused attention implementations. On the other hand, the gradient-based metrics require retaining all activations until the backward pass to compute gradients, which fundamentally prevents early activation eviction during the forward pass and thus negates activation memory savings. Together, these limi-

tations motivate the need for a lightweight, forward-only importance metric.

**Proposed $\ell_2$-Norm Criterion.** We propose an importance criterion motivated by the residual structure of LLMs. LLMs are composed of a stack of Transformer layers (Vaswani et al., 2017), each composed of an MHA block followed by an FFN block. These blocks update the input feature through an additive structure as follows:

$$\mathbf{x}_{l+1} = \mathbf{x}_l + f_l(\mathbf{x}_l), \tag{5}$$

where $f_l(\cdot)$ denotes either MHA or FFN at depth $l$, and $\mathbf{x}_l$ denotes the residual stream input to $f_l$. Under this formulation, $f_l(\mathbf{x}_l)$ represents the incremental feature contribution of block $l$ added to the residual stream $\mathbf{x}_l$ (Elhage et al., 2021). We hypothesize that since these blocks affect downstream representations solely through additive updates, the magnitude of a token's residual update reflects its contribution to the subsequent computation and, consequently, training. Accordingly, we define the importance score of token $t$ at block $l$ as the $\ell_2$-norm of its residual update, as below:

$$I_{l,t} = \|f_l(\mathbf{x}_{l,t})\|_2. \tag{6}$$

Here, $f_l(\mathbf{x}_{l,t}) \in \mathbb{R}^h$ denotes the output vector of a block at depth $l$ corresponding to token $t$, where $h$ is the hidden dimension.

The proposed criterion can be computed efficiently during the forward pass with negligible overhead, without interfering with fused attention kernels or requiring gradient information.

### 3.3. Token Selection Strategy

Given the token-level importance scores defined in Section 3.2, a natural baseline is to perform token selection at each block by retaining top-$k$ tokens using only the importance scores computed at that block. While straightforward, such locally driven selection fails to account for cross-block token contributions and is therefore insufficient for stable and effective fine-tuning, as discussed in detail below.

#### 3.3.1. LIMITATIONS OF LOCAL SCORE-BASED SELECTION

Local score-based selection makes retention decisions using only instantaneous importance scores at each block. Since residual updates accumulate across depth, such local decisions can lead to inconsistent token sets across blocks. Figure 3 visualizes the pairwise Jaccard similarity between the top-50% token sets selected at different blocks. While token sets from adjacent blocks exhibit high similarity, indicating strong local consistency, the similarity between distant blocks is substantially lower. This inconsistency implies that gradients for tokens dropped at later blocks must

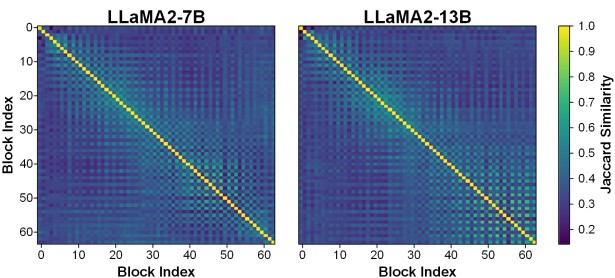

*Figure 3.* Jaccard similarity between the top-50% token sets selected by the proposed $\ell_2$-norm importance scores across blocks. Results are reported on MMLU using LLaMA2-7B and LLaMA2-13B.

still propagate backward through earlier blocks along the residual paths, resulting in misaligned gradient flow and potential training instability. Furthermore, importance scores computed at earlier blocks lack the foresight to account for updates at later ones. Consequently, tokens deemed important based on early-block scores may ultimately contribute negligibly to the overall training objective, leading to suboptimal performance. To address the limitations, we introduce two techniques: cumulative token selection (Section 3.3.2) and lazy selection scheduling (Section 3.3.3).

### 3.3.2. CUMULATIVE TOKEN SELECTION

To tackle the instability caused by locally driven token selection, we introduce cumulative token selection, a structured retention strategy that enforces monotonicity of token sets across depth. Specifically, cumulative token selection progressively increases the token retention ratio across blocks and imposes that any token retained at a block is deterministically **retained at all subsequent ones**. This design results in the inclusion relationship as follows:

$$\mathcal{S}_1 \subseteq \mathcal{S}_2 \subseteq \cdots \subseteq \mathcal{S}_L, \tag{7}$$

where $\mathcal{S}_l$ denotes the set of tokens retained at block $l$.

This nested structure is pivotal for maintaining training stability. Under this scheme, gradients of tokens dropped at a certain block never propagate to preceding blocks. This ensures that for any retained token, the gradient always flows through both the non-linear transformation and the residual bypass of each block, up to the block where it is eventually dropped. This prevents degenerate cases where gradients flow only through the residual path. By avoiding such *partially corrupted* gradient paths, our approach preserves gradient consistency and stabilizes training dynamics.

### 3.3.3. LAZY SELECTION SCHEDULING

While cumulative token selection significantly enhances training stability by ensuring backward path consistency, its effectiveness is still constrained by the locality of block-wise

---

**Algorithm 1** Lazy Selection Scheduling

**Input:** activation memory cost $\{M_l\}_{l=1}^L$, target activation memory budget $\mathcal{B}$, memory margin $\alpha$, retention token counts $\{k_l\}_{l=1}^L$, complete token set $\mathcal{T} = \{1, \ldots, T\}$
**Output:** retained token set $\{\mathcal{S}_l\}_{l=1}^L$, activation cache $\mathcal{C}$
**Initialization:**
1: $\mathcal{C} \leftarrow \emptyset$          // activation cache for backward
2: $\mathcal{G} \leftarrow \emptyset$        // group of blocks awaiting selection
3: $\tilde{I} \leftarrow \mathbf{0}$    // accumulated token-wise importance over $\mathcal{G}$
4: $\mathcal{M} \leftarrow 0$       // current activation memory usage of $\mathcal{C}$
**Forward of LLM:**
5: **for** $l = 1, \ldots, L$ **do**
6:      $\mathcal{G} \leftarrow \mathcal{G} \cup \{l\}$
7:      $(\mathcal{A}_l, \mathcal{C}) \leftarrow \text{FORWARD}(l, \mathcal{C})$     // $\mathcal{A}_l$: block output.
8:      $\tilde{I} \leftarrow \tilde{I} + \text{SCORE}(\mathcal{A}_l)$
9:      $\mathcal{M} \leftarrow \mathcal{M} + M_l$
10:     // *Check whether the memory budget is near limit.*
11:     **if** $\mathcal{M} + \alpha > \mathcal{B}$ **then**
12:         **for** $j \in \mathcal{G}$ **do**
13:            $\mathcal{S}_j \leftarrow \text{TOPK}(\tilde{I}, k_j)$
14:            $\mathcal{C} \leftarrow \text{EVICTACTIVATION}(\mathcal{C}, j, \mathcal{T} \setminus \mathcal{S}_j)$
15:            $\mathcal{M} \leftarrow \mathcal{M} - \frac{T-k_j}{T} \times M_j$
16:         **end for**
17:         $\mathcal{G} \leftarrow \emptyset; \ \tilde{I} \leftarrow \mathbf{0}$
18:     **end if**
19: **end for**

---

importance scores, which capture only partial evidence of a token's overall contribution. To overcome this, we introduce lazy selection scheduling that defers token selection and the subsequent eviction of non-informative activations to a later stage, as detailed in Algorithm 1. Instead of eagerly discarding tokens at every block based on local scores, lazy scheduling triggers the selection and eviction only when the activation memory budget is nearing its limit, enabling the aggregation of importance scores across multiple blocks. Formally, we partition the blocks into groups, where each block within a group shares a common selection trigger. Within each group $g$, the accumulated importance score of token $t$ is defined as follows:

$$\widetilde{I}_{g,t} = \sum_{l \in g} I_{l,t} \tag{8}$$

where $I_{l,t}$ is the token-level score at depth $l$ defined in Eq. (6). Note that token selection is still performed in a block-wise manner by retaining the top-$k$ tokens according to $\widetilde{I}_{g,t}$. At the group boundary, only the activations corresponding to selected tokens are kept in the memory, whereas the activations of non-selected tokens are discarded. By combining cumulative token selection with delayed selection based on accumulated importance scores, TokenDrop enables globally informed token retention while maintaining stable training dynamics under activation memory budgets.

## 3.4. Sensitivity-Aware Retention Ratio Allocation

While Sections 3.2 and 3.3 define how to identify informative tokens across blocks, it remains to determine how many tokens should be retained at each block. To this end, we adopt a sensitivity-aware allocation strategy that assigns block-wise token retention ratios according to each block's sensitivity to backward skipping. Similar to the sensitivity calculation scheme in Woo et al. (2024b), sensitivity is estimated based on the deviation of weight gradients induced by backward skip. Specifically, using a small batch of samples, we perform a single diagnostic iteration in which a fixed fraction of tokens is uniformly skipped during backpropagation across all blocks, and measure the resulting weight gradient deviation at each block. Under cumulative token selection, gradients of tokens dropped at deeper blocks do not propagate into earlier blocks. Therefore, the impact of token dropping at block $l$ must account for gradient deviations accumulated over all preceding blocks. Accordingly, we define the sensitivity of block $l$ as the aggregated gradient deviation up to that depth:

$$Sensitivity^{(l)} = \sum_{i=1}^{l} \left\| \mathbf{G}_W^{(i)} - \widetilde{\mathbf{G}}_W^{(i)} \right\|_2 \qquad (9)$$

where $\mathbf{G}_W^{(i)}$ and $\widetilde{\mathbf{G}}_W^{(i)}$ denote the weight gradients of block $i$ computed with full and sparsified backpropagation, respectively. Based on the estimated sensitivities, we employ a greedy allocation, following Liu et al. (2022); Woo et al. (2024a;b), that progressively lowers the retention ratio of the least sensitive blocks until the target FLOP constraint is satisfied. Given the depth-wise increase in sensitivity, this process ensures monotonically non-decreasing token retention ratios across blocks, satisfying Eq. (7). Details of the allocation strategy are provided in Appendix B.

## 4. Experiments

To demonstrate the effectiveness of TokenDrop, we conducted extensive experiments across diverse LLM fine-tuning scenarios, including **multitask language understanding**, **commonsense reasoning**, **instruction following**, **mathematical reasoning**, and **code generation**. We compared TokenDrop against two representative backpropagation skipping approaches, DropBP (Woo et al., 2024b) and TokenTune (Simoulin et al., 2024), in terms of memory usage, training speedup, and accuracy. For efficient fine-tuning, we employed FlashAttention-2 (Dao, 2024) and a padding-free `DataCollator` in PyTorch-based training pipeline. Since the original TokenTune implementation is not compatible with these optimizations, we re-implemented TokenTune within our framework for a fair comparison. LoRA (Hu et al., 2022) was adopted as the default fine-tuning configuration. To further evaluate the generality of our approach, Section 4.3 extends the evaluation to full

fine-tuning, QLoRA (Dettmers et al., 2023), and an adapter-free PEFT method, PaCA (Woo et al., 2025). All experiments were conducted on a single NVIDIA A100 80GB GPU (Choquette et al., 2021). In addition, we conducted each experiment three times with different seeds and reported the average accuracy with standard deviation. Detailed hyperparameter settings are provided in Appendix G.

### 4.1. Experimental Setup

**Multitask Language Understanding.** We fine-tuned LLaMA2-7B and LLaMA2-13B (Touvron et al., 2023) on the Massive Multitask Language Understanding (MMLU) training dataset (Hendrycks et al., 2020) and reported 5-shot accuracies using *lm-evaluation-harness* (Biderman et al., 2024).

**Commonsense Reasoning.** We fine-tuned LLaMA2-7B and LLaMA2-13B on 8 commonsense benchmarks (CSR170K): ARC-Challenge, ARC-Easy (Clark et al., 2018), BoolQ (Clark et al., 2019), HellaSwag (Zellers et al., 2019), OpenBookQA (Mihaylov et al., 2018), PIQA (Bisk et al., 2020), SIQA (Sap et al., 2019), and WinoGrande (Sakaguchi et al., 2021). Evaluation was conducted in a 0-shot setting via *lm-evaluation-harness*.

**Instruction Following.** To evaluate the impact of TokenDrop on instruction-following capabilities, we fine-tuned LLaMA3.1-8B (Dubey et al., 2024) on the OpenAssistant-Guanaco dataset (Köpf et al., 2023; Dettmers et al., 2023). We then evaluated the model using MT-Bench, a benchmark designed to measure multi-turn instruction-following and conversational quality (Zheng et al., 2023). GPT-4o-mini (Achiam et al., 2023) was employed as a judge to evaluate model responses on MT-Bench.

**Mathematical Reasoning.** We fine-tuned LLaMA3.1-8B on the MetaMathQA-40K dataset (Yu et al., 2024) and evaluated it on GSM8K (Cobbe et al., 2021) using *lm-evaluation-harness* with chain-of-thought prompting.

**Code Generation.** We fine-tuned LLaMA3.1-8B on Evol-Instruction-66K (CodeFuse AI, 2023) and evaluated Pass@1 accuracy on HumanEval+ (Liu et al., 2023) with *EvalPlus* (Liu et al., 2023) using greedy decoding and temperature set to zero.

### 4.2. Main Results

**Results on Multitask Language Understanding.** As shown in Table 10, TokenDrop achieves accuracies comparable to the baseline LoRA while substantially reducing both memory footprint and fine-tuning time. Specifically, when fine-tuning LLaMA2-7B, TokenDrop reduces memory by 41.7% and accelerates fine-tuning by 1.35×, while even

*Table 1.* Comparison of peak memory usage, training speedup, and 5-shot MMLU accuracy for LLaMA2-7B and LLaMA2-13B fine-tuning using different backpropagation skipping methods.

| Model | Method | Peak Memory | SpeedUp | Accuracy (%) | | | | |
|---|---|---|---|---|---|---|---|---|
| | | | | Hum. | STEM | Social | Other | Avg. |
| | Not Tuned | - | - | 43.3 | 37.1 | 51.8 | 52.8 | 45.9 |
| **LLaMA2-7B** | LoRA | 63.9 GB | 1.00× | 50.5±0.4 | 42.4±0.6 | 61.4±0.6 | 59.5±0.5 | 53.1±0.2 |
| | LoRA+DropBP | 49.9 GB | 1.29× | 48.0±1.2 | 41.3±0.6 | 58.4±1.4 | 58.1±1.1 | 51.0±0.7 |
| | LoRA+TokenTune | 38.8 GB | 1.31× | 46.4±1.4 | 40.8±0.4 | 57.9±0.2 | 57.8±0.2 | 50.1±0.5 |
| | LoRA+TokenDrop | **37.3 GB** | **1.35×** | 51.2±0.2 | 43.7±1.2 | 62.7±0.5 | 60.3±0.4 | **54.1±0.3** |
| | Not Tuned | - | - | 53.1 | 44.2 | 62.8 | 60.8 | 54.9 |
| **LLaMA2-13B** | LoRA | 66.8 GB | 1.00× | 55.8±0.2 | 47.1±0.5 | 66.8±0.6 | 64.3±0.5 | **58.1±0.4** |
| | LoRA+DropBP | 54.9 GB | **1.28×** | 53.4±2.5 | 45.8±0.6 | 64.1±1.2 | 62.1±1.5 | 56.0±1.5 |
| | LoRA+TokenTune | **50.4 GB** | 1.19× | 52.8±0.9 | 46.0±1.1 | 65.3±0.4 | 63.2±0.6 | 56.3±0.3 |
| | LoRA+TokenDrop | 50.8 GB | **1.28×** | 55.9±0.8 | 46.7±0.2 | 66.1±0.7 | 64.1±0.4 | 57.9±0.4 |

*Table 2.* Comparison of peak memory usage, training speedup, and 0-shot accuracy on commonsense reasoning benchmarks when fine-tuning LLaMA2-7B and LLaMA2-13B with different backpropagation skipping methods.

| Model | Method | Peak Memory | SpeedUp | Accuracy (%) | | | | | | | | |
|---|---|---|---|---|---|---|---|---|---|---|---|---|
| | | | | ARC-c | ARC-e | BoolQ | Hella | OBQA | PIQA | SIQA | Wino | Avg. |
| | Not Tuned | - | - | 46.2 | 74.6 | 77.7 | 76.0 | 44.2 | 79.1 | 46.1 | 69.1 | 64.1 |
| **LLaMA2-7B** | LoRA | 44.9 GB | 1.00× | 55.2±0.7 | 82.5±0.7 | 79.6±2.4 | 80.0±0.1 | 51.3±0.6 | 81.8±0.3 | 59.0±0.2 | 83.1±0.4 | **71.6±0.4** |
| | LoRA+DropBP | 35.7 GB | 1.21× | 53.5±0.9 | 80.9±0.4 | 79.8±5.2 | 79.3±0.2 | 47.9±1.4 | 80.9±0.4 | 57.4±0.2 | 80.5±0.7 | 70.0±0.8 |
| | LoRA+TokenTune | 33.4 GB | 1.18× | 53.7±0.5 | 81.9±0.2 | 80.6±2.7 | 79.0±0.3 | 47.5±0.5 | 81.1±0.3 | 58.1±0.4 | 82.0±0.7 | 70.5±0.2 |
| | LoRA+TokenDrop | **32.9 GB** | **1.23×** | 54.0±0.8 | 81.5±0.4 | 85.2±0.9 | 79.0±0.2 | 49.2±0.2 | 81.0±0.1 | 58.1±0.2 | 82.2±0.4 | 71.3±0.1 |
| | Not Tuned | - | - | 49.0 | 77.4 | 80.6 | 79.4 | 45.2 | 80.5 | 47.4 | 72.2 | 66.5 |
| **LLaMA2-13B** | LoRA | 65.2 GB | 1.00× | 61.5±0.4 | 85.1±0.5 | 84.4±1.2 | 82.8±0.1 | 55.2±0.7 | 83.5±0.2 | 61.1±0.3 | 85.4±0.4 | **74.9±0.2** |
| | LoRA+DropBP | 54.8 GB | **1.30×** | 60.9±0.3 | 84.4±0.2 | 83.4±2.9 | 82.5±0.1 | 51.1±0.1 | 83.1±0.3 | 59.8±0.4 | 83.7±0.5 | 73.6±0.4 |
| | LoRA+TokenTune | 51.9 GB | 1.25× | 60.5±1.2 | 84.8±0.2 | 83.3±1.9 | 82.3±0.4 | 51.3±2.0 | 83.0±0.2 | 60.1±0.3 | 85.4±0.8 | 73.8±0.7 |
| | LoRA+TokenDrop | **51.6 GB** | **1.30×** | 60.5±0.2 | 84.6±0.5 | 87.5±0.9 | 82.2±0.1 | 52.3±0.3 | 82.7±0.2 | 60.5±0.4 | 84.6±0.5 | 74.4±0.2 |

yielding a notable improvement in accuracy. This result suggests that filtering uninformative tokens not just reduces training overhead but also enhances model performance, consistent with observations in prior work (Lin et al., 2024; Pang et al., 2025). On the other hand, DropBP and To-kenTune incur a noticeable accuracy degradation compared to standard LoRA, and TokenDrop delivers a clear performance gain over those conventional approaches, achieving up to 25.3% greater peak memory usage reduction and up to 4.7% faster training speed. For LLaMA2-13B, TokenDrop reduces the memory footprint by 23.9% and accelerates training by 28%, incurring negligible accuracy degradation. Moreover, TokenDrop consistently outperforms other backpropagation skipping methods in accuracy, while exhibiting comparable or superior training efficiency.

**Results on Commonsense Reasoning.** Table 11 reports the results on commonsense reasoning benchmarks. For LLaMA2-7B, TokenDrop reduces peak memory usage by 26.7% and fine-tuning time by 18.7%, while maintaining a closely matching average accuracy. In addition, TokenDrop achieves the highest average accuracy among all compared skipping methods while using the least memory and training time. A similar trend is consistently observed for LLaMA2-13B. TokenDrop attains a 20.9% reduction in memory usage and a 1.30× speedup with a relatively small accuracy drop.

**Results on Instruction Following.** Table 3 shows the results on instruction following performance evaluated using MT-Bench, where scores are computed by averaging the Turn 1 and Turn 2 results. When fine-tuning LLaMA3.1-8B, TokenDrop significantly decreases memory consumption and training time without sacrificing task performance, and in some tasks, achieves improved scores. In particular, TokenDrop reduces peak memory usage by 42.9% and achieves a 1.38× wall-clock speedup compared to the LoRA baseline, while attaining the highest average MT-Bench score.

**Results on Mathematical Reasoning.** Table 4 reports the results on GSM8K with chain-of-thought prompting. TokenDrop substantially reduces training cost while preserving mathematical reasoning performance. Compared to the LoRA baseline, TokenDrop reduces peak memory usage from 54.2 GB to 38.2 GB, corresponding to a 29.5% reduction, and achieves a 1.27× training speedup. At the same time, TokenDrop maintains the same accuracy as LoRA,

*Table 3.* Comparison of peak memory usage, training speedup, and MT-Bench score when fine-tuning LLaMA3.1-8B on OpenAssistant-Guanaco.

| Model | Method | Memory | SpeedUp | Writing | Role. | Reason. | Math | Coding | Extract. | STEM | Human. | Avg. |
|---|---|---|---|---|---|---|---|---|---|---|---|---|
| | Not Tuned | - | - | 5.35 | 5.20 | 2.55 | 2.75 | 2.00 | 2.70 | 5.40 | 3.60 | 3.69 |
| | LoRA | 78.0 GB | 1.00× | 5.55±0.48 | 5.70±0.43 | 3.25±0.18 | 2.62±0.36 | 2.90±0.13 | 4.52±0.51 | 6.10±0.17 | 6.50±0.15 | 4.64±0.16 |
| **LLaMA3.1-8B** | LoRA+DropBP | 48.1 GB | 1.30× | 4.95±0.74 | 5.08±0.53 | 3.40±0.59 | 2.45±0.30 | 2.68±0.25 | 3.43±0.88 | 5.73±0.58 | 5.80±0.28 | 4.19±0.26 |
| | LoRA+TokenTune | 45.7 GB | 1.27× | 4.15±1.21 | 4.42±0.38 | 2.43±0.03 | 2.48±0.63 | 2.35±0.00 | 3.18±0.53 | 5.42±0.40 | 4.87±1.01 | 3.66±0.32 |
| | LoRA+TokenDrop | **44.6 GB** | **1.38×** | 5.73±0.13 | 5.38±0.14 | 3.13±0.10 | 2.95±0.20 | 2.93±0.19 | 4.65±0.82 | 6.02±0.16 | 6.45±0.05 | **4.66±0.13** |

achieving 54.9% accuracy. In contrast, DropBP and TokenTune show a noticeable accuracy degradation, while providing smaller memory and speed improvements than TokenDrop. These results indicate that TokenDrop can effectively reduce fine-tuning overhead even for complicated reasoning tasks without compromising final performance.

*Table 4.* Comparison of peak memory usage, training speedup, and accuracy on GSM8K when fine-tuning LLaMA3.1-8B on MetaMathQA-40K and evaluating with chain-of-thought prompting.

| Method | Memory | SpeedUp | Accuracy (%) |
|---|---|---|---|
| Not Tuned | - | - | 53.4 |
| LoRA | 54.2 GB | 1.00× | **54.9±1.8** |
| LoRA+DropBP | 44.7 GB | 1.21× | 48.6±0.7 |
| LoRA+TokenTune | 40.4 GB | 1.19× | 53.8±2.8 |
| LoRA+TokenDrop | **38.2 GB** | **1.27×** | **54.9±1.0** |

**Results on Code Generation.** Table 5 shows the Pass@1 results when evaluating LLaMA3.1-8B on the HumanEval+ benchmark. TokenDrop consistently provides a favorable trade-off between efficiency and accuracy. Compared with LoRA, TokenDrop yields a 28.9% memory reduction, and accelerates fine-tuning by 1.24×. Despite this reduction in training cost, TokenDrop closely matches the LoRA baseline in Pass@1 accuracy, achieving 36.6% compared to 36.8%. Notably, TokenDrop outperforms both DropBP and TokenTune in accuracy while using less memory. These results further highlight the effectiveness of importance-aware token-level backward skipping in preserving code generation performance under reduced memory and computational costs.

*Table 5.* Comparison of peak memory usage, training speedup, and Pass@1 accuracy on the HumanEval+ benchmark when fine-tuning LLaMA3.1-8B on Evol-Instruction-66K.

| Method | Memory | SpeedUp | Pass@1 (%) |
|---|---|---|---|
| Not Tuned | - | - | 30.5 |
| LoRA | 66.8 GB | 1.00× | **36.8±1.2** |
| LoRA+DropBP | 53.5 GB | **1.24×** | 35.0±1.1 |
| LoRA+TokenTune | 50.9 GB | 1.16× | 35.4±0.9 |
| LoRA+TokenDrop | **47.5 GB** | **1.24×** | 36.6±0.5 |

### 4.3. Compatibility with Full Fine-Tuning and Other PEFT Methods

To demonstrate the compatibility of TokenDrop with full fine-tuning or other PEFT methods, we applied TokenDrop to full fine-tuning, QLoRA (Dettmers et al., 2023), and PaCA (Woo et al., 2025). QLoRA reduces memory consumption by quantizing pretrained weights into low-bit representations, while PaCA adopts an adapter-free design that fine-tunes only a subset of randomly selected output channels of pretrained weights. TokenDrop is orthogonal to these methods and can be applied on top of them to further reduce activation memory consumption and training time.

As shown in Table 6, TokenDrop consistently achieves additional memory and training time reductions across different fine-tuning algorithms. Notably, when applied to full fine-tuning, TokenDrop leads to a substantial improvement in accuracy alongside a 1.50× speedup. When applied to QLoRA and PaCA, TokenDrop decreases peak memory usage by up to 41.4% and accelerates training by up to 1.32×, without compromising accuracy.

### 4.4. Impact of Cumulative Token Selection and Lazy Selection Scheduling

Table 7 presents the results of an ablation study on LLaMA2-7B, examining the effects of cumulative token selection and lazy selection scheduling. Here, the baseline represents token selection with a block-wise local importance score. Introducing cumulative token selection leads to performance gains on both MMLU and CSR170K, and is particularly critical for stabilizing training on MMLU. This result indicates that preserving gradient integrity across depth is crucial for stabilizing training under token-level skipping. In addition, lazy selection scheduling further improves performance, highlighting the benefit of deferring token selection to incorporate importance signals accumulated across multiple blocks.

### 4.5. Comparisons of Token Importance Criteria

Table 8 compares different token importance criteria for token selection: *Random*, $\ell_2$-*Norm-Small*, and $\ell_2$-*Norm-Large* (ours). *Random* selection serves as an importance-agnostic baseline, uniformly retaining tokens regardless of

*Table 6.* Compatibility of TokenDrop with full fine-tuning and different PEFT methods. Results report memory usage, training speedup, and 5-shot MMLU accuracy for LLaMA models.

| Model | Method | Peak Memory | SpeedUp | Accuracy (%) | | | | |
|---|---|---|---|---|---|---|---|---|
| | | | | Hum. | STEM | Social | Other | Avg. |
| **LLaMA3.2-1B** | Not Tuned | - | - | 28.8 | 28.1 | 32.1 | 36.0 | 31.0 |
| | Full-FT | 66.8 GB | 1.00× | 38.8±0.5 | 35.0±0.6 | 46.1±2.2 | 44.9±0.2 | 40.9±0.7 |
| | Full-FT+TokenDrop | **47.4 GB** | **1.50×** | 38.8±0.6 | 38.5±1.0 | 46.3±0.8 | 46.9±0.3 | **42.2±0.5** |
| **LLaMA2-7B** | Not Tuned | - | - | 43.3 | 37.1 | 51.8 | 52.8 | 45.9 |
| | QLoRA | 54.8 GB | 1.00× | 49.2±0.7 | 42.2±0.8 | 61.1±1.3 | 58.9±0.3 | 52.4±0.7 |
| | QLoRA+TokenDrop | **32.1 GB** | **1.32×** | 50.2±0.9 | 42.4±0.4 | 60.8±0.6 | 59.1±0.3 | **52.7±0.4** |
| | PaCA | 54.3 GB | 1.00× | 49.2±1.1 | 42.2±0.9 | 61.1±0.2 | 58.9±0.4 | 52.4±0.6 |
| | PaCA+TokenDrop | **37.2 GB** | **1.31×** | 49.3±1.2 | 42.2±0.5 | 60.8±0.4 | 59.5±0.5 | **52.5±0.5** |

*Table 7.* Ablation study results of 5-shot MMLU accuracy and 0-shot commonsense reasoning accuracy when fine-tuning LLaMA2-7B.

| Method | MMLU Acc. (%) | $\Delta$ | CSR170K Acc. (%) | $\Delta$ |
|---|---|---|---|---|
| Baseline | 40.1 | - | 69.7 | - |
| + Cumulative Selection | 53.3 | +13.2 | 70.2 | +0.5 |
| + Lazy Scheduling | **54.1** | +0.8 | **71.3** | +1.1 |

their contribution. $\ell_2$-*Norm-Small* retains tokens with the smallest norms of residual update, while $\ell_2$-*Norm-Large* selects tokens with the largest ones. In both MMLU and CSR170K, $\ell_2$-*Norm-Large* consistently outperforms the alternative criteria. Compared to *Random* selection, it improves MMLU accuracy by 3.7 points and CSR170K accuracy by 1.0 point. In contrast, $\ell_2$-*Norm-Small* results in noticeable performance degradation, even underperforming the random baseline. This observation suggests that retaining tokens with weak residual signals provides little useful information and hinders effective learning. Together, these results validate the magnitude of residual updates as a robust and reliable indicator of token importance for fine-tuning.

*Table 8.* Comparison of 5-shot MMLU accuracy and 0-shot commonsense reasoning accuracy across various token importance criteria when fine-tuning LLaMA2-7B.

| Method | MMLU Acc. (%) | CSR170K Acc. (%) |
|---|---|---|
| Random | 50.4 | 70.3 |
| $\ell_2$-Norm-Small | 47.0 | 67.9 |
| $\ell_2$-Norm-Large (Ours) | **54.1** | **71.3** |

## 5. Conclusion

In this work, we introduce TokenDrop, which enables efficient LLM fine-tuning by selectively skipping backpropagation at the token level based on importance signals. TokenDrop integrates residual-based importance estimation with cumulative and delayed selection strategies to substantially reduce activation memory and training time while maintaining accuracy. Across diverse fine-tuning settings, TokenDrop achieves up to 42.9% reduction in total memory usage and up to 1.50× wall-clock speedup while maintaining downstream task performance.

## Acknowledgement

This work was supported in part by the Institute of Information and Communications Technology Planning and Evaluation under Grant RS-2025-02218733, Grant RS-2021-II211343, Grant IITP-2025-RS-2023-00256081, and Grant RS-2024-00347394, and in part by the National Research Foundation of Korea under Grant RS-2024-00408040 and Grant RS-2022-00144419.

## Impact Statement

This paper presents work whose goal is to advance the field of Machine Learning. There are many potential societal consequences of our work, none of which we feel must be specifically highlighted here.

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

# A. Gradient Skipping in Self-Attention

This section analyzes the effect of skipping gradients with respect to the keys and values of dropped tokens in self-attention. We show that although these gradients are generally non-zero, their contribution is negligible and can be safely skipped in practice.

## A.1. Forward Computation of Self-Attention

For brevity, we consider a self-attention block with a single head. The forward computation of the attention block is given as follows:

$$Q = XW_Q^\top, \quad K = XW_K^\top, \quad V = XW_V^\top, \tag{10}$$

$$O = \frac{QK^\top}{\sqrt{d}} + M, \qquad P = \text{softmax}(O), \tag{11}$$

$$Y = PV, \tag{12}$$

where $X \in \mathbb{R}^{T \times d}$ denotes the input to the attention block with sequence length $T$ and hidden dimension $d$, $Q, K, V \in \mathbb{R}^{T \times d}$ are the query, key, and value, respectively, and $W_Q, W_K, W_V \in \mathbb{R}^{d \times d}$ are the corresponding projection weight matrices. $O$, $P \in \mathbb{R}^{T \times T}$ represent the attention scores before and after the softmax operation, and $M \in \mathbb{R}^{T \times T}$ is the causal attention mask.

## A.2. Consequences of Row-Wise Gradient Skipping

Let $G_Y = \partial\mathcal{L}/\partial Y$ denotes the upstream gradient and $\mathcal{S} \subseteq \{1, \ldots, T\}$ denotes the retained token set. For all $i \notin \mathcal{S}$, we have

$$(G_Y)_{i,:} = \mathbf{0}, \tag{13}$$

indicating that backpropagation is skipped for non-retained tokens.

From $Y = PV$, the gradients with respect to $P$ and $V$ are

$$\frac{\partial\mathcal{L}}{\partial P} = G_Y V^\top, \qquad \frac{\partial\mathcal{L}}{\partial V} = P^\top G_Y. \tag{14}$$

Thus, $(\partial\mathcal{L}/\partial P)_{i,:} = \mathbf{0}$ for all $i \notin \mathcal{S}$. Since the softmax is applied row-wise, the corresponding score gradients also satisfy

$$\left(\frac{\partial\mathcal{L}}{\partial O}\right)_{i,:} = \mathbf{0} \quad \forall i \notin \mathcal{S}. \tag{15}$$

With $O = QK^\top/\sqrt{d} + M$, the gradients with respect to $Q$ and $K$ are

$$\frac{\partial\mathcal{L}}{\partial Q} = \frac{1}{\sqrt{d}}\frac{\partial\mathcal{L}}{\partial O}K, \qquad \frac{\partial\mathcal{L}}{\partial K} = \frac{1}{\sqrt{d}}\left(\frac{\partial\mathcal{L}}{\partial O}\right)^\top Q. \tag{16}$$

As a consequence,

$$\left(\frac{\partial\mathcal{L}}{\partial Q}\right)_{i,:} = \mathbf{0} \quad \forall i \notin \mathcal{S}, \tag{17}$$

whereas each gradient with respect to the keys of dropped tokens generally remains

$$\left(\frac{\partial\mathcal{L}}{\partial K}\right)_{i,:} = \frac{1}{\sqrt{d}}\sum_{j \in \mathcal{S}}\left(\frac{\partial\mathcal{L}}{\partial O}\right)_{ji} Q_{j,:}, \tag{18}$$

which is non-zero whenever retained queries exist.

Similarly, for the value path,

$$\left(\frac{\partial\mathcal{L}}{\partial V}\right)_{i,:} = \sum_{j \in \mathcal{S}} P_{ji}(G_Y)_{j,:}, \tag{19}$$

implying that value gradients of dropped tokens are also generally non-zero.

### A.3. Impact of Discarding Key and Value Gradients for Dropped Tokens

As discussed above, skipping backpropagation for a subset of tokens does not eliminate their gradients with respect to keys and values, due to the cross-token dependencies inherent in the attention mechanism. Consequently, without further approximation, gradient computations for $X, W_K, W_V$ cannot fully leverage token-level gradient skipping and require a fully stored attention input $X$ as below:

$$\frac{\partial \mathcal{L}}{\partial X} = \frac{\partial \mathcal{L}}{\partial Q_{\mathcal{S}}} W_Q + \frac{\partial \mathcal{L}}{\partial K} W_K + \frac{\partial \mathcal{L}}{\partial V} W_V, \tag{20}$$

$$\frac{\partial \mathcal{L}}{\partial W_K} = \left( \frac{\partial \mathcal{L}}{\partial K} \right)^\top X, \qquad \frac{\partial \mathcal{L}}{\partial W_V} = \left( \frac{\partial \mathcal{L}}{\partial V} \right)^\top X. \tag{21}$$

Here, we exceptionally use $Q_{\mathcal{S}} \in \mathbb{R}^{T \times d}$ to denote a masked query tensor where entries corresponding to dropped tokens $i \notin \mathcal{S}$ are explicitly zeroed out (i.e., not $Q_{\mathcal{S}} \in \mathbb{R}^{|S| \times d}$).

Importantly, under the cumulative token selection strategy, gradients of dropped tokens are never required by preceding blocks during backpropagation. Therefore, although the exact expression of $\partial \mathcal{L}/\partial X$ includes contributions from $\partial \mathcal{L}/\partial K_i$ and $\partial \mathcal{L}/\partial V_i$ for dropped tokens, these terms can be safely omitted in TokenDrop. As a result, the remaining concern lies in $\partial \mathcal{L}/\partial W_K$ and $\partial \mathcal{L}/\partial W_V$.

To avoid storing the full input activations and performing weight gradient computations for dropped tokens, TokenDrop explicitly discards the gradients $\partial \mathcal{L}/\partial K_i$ and $\partial \mathcal{L}/\partial V_i$ for $i \notin \mathcal{S}$. Accordingly, $W_K$ and $W_V$ are updated using only the retained tokens as follows:

$$\frac{\partial \mathcal{L}}{\partial W_K} = \left( \frac{\partial \mathcal{L}}{\partial K_{\mathcal{S}}} \right)^\top X_{\mathcal{S}}, \qquad \frac{\partial \mathcal{L}}{\partial W_V} = \left( \frac{\partial \mathcal{L}}{\partial V_{\mathcal{S}}} \right)^\top X_{\mathcal{S}}. \tag{22}$$

To assess the practical impact of this approximation, we directly compare downstream training performance between (i) updating $W_K$ and $W_V$ using all tokens with the full key and value gradients, and (ii) updating them using only the retained tokens by omitting the gradients of dropped tokens, as in TokenDrop. As presented in Table 9, we observe no degradation in average accuracy. These results indicate that explicitly zeroing $\partial \mathcal{L}/\partial K_i$ and $\partial \mathcal{L}/\partial V_i$ for dropped tokens enables simplified backward computation while preserving training effectiveness.

*Table 9.* Effect of discarding key and value gradients for dropped tokens on 5-shot MMLU accuracy when fine-tuning LLaMA2-7B.

| Method | Hums. | STEM | Social. | Other | Avg. |
|---|---|---|---|---|---|
| $W_K, W_V$ Update with Full Tokens | 51.9 | 44.2 | 61.0 | 60.6 | **54.1** |
| $W_K, W_V$ Update with Only Retained Tokens (Ours) | 51.2 | 43.7 | 62.7 | 60.3 | **54.1** |

## B. Block-wise Retention Ratio Allocation Algorithm

This section provides implementation details of the sensitivity-aware retention ratio allocation strategy described in Section 3.4. Specifically, we detail the greedy procedure used to assign block-wise token retention ratios under a global backward FLOPs constraint, following prior work (Liu et al., 2022; Woo et al., 2024a;b). The goal is to satisfy a target computational budget while retaining informative tokens as much as possible, as formulated below:

$$
\begin{aligned}
\max_{\{r_l\}} \quad & \sum_{l=1}^{L} r_l \\
\text{s.t.} \quad & \sum_{l=1}^{L} F_l \times r_l \leq F_{\text{target}}, \\
& r_l \in [0, 1], \quad \forall l,
\end{aligned}
\tag{23}
$$

where $r_l$ denotes the retention ratio of each block $l \in \{1, \ldots, L\}$, $F_l$ denotes the backward FLOPs of block at depth $l$, and $F_{\text{target}}$ is the target total backward FLOPs. A detailed algorithm to solve this objective is described in Algorithm 2. We employ a heap-based greedy allocation strategy that progressively assigns a lower retention ratio to a less sensitive block until the total FLOPs constraint is satisfied. At each step, the algorithm selects the block with the smallest $S_l \times (1 - r_l)$ value and reduces its retention ratio by $\Delta r$, updating the total FLOPs accordingly. This process continues until the total FLOP count is no greater than $F_{\text{target}}$.

---

**Algorithm 2** Block-wise Retention Ratio Allocation under backward FLOPs Constraint

---

1: **Input:** FLOPs $\{F_l\}_{l=1}^{L}$, sensitivities $\{S_l\}_{l=1}^{L}$, target FLOPs $F_{\text{target}}$, retention ratio step size $\Delta r$
2: **Output:** retention ratios $\{r_l\}_{l=1}^{L}$
3: Initialize $r_l \leftarrow 1$ for all $l$
4: Compute total FLOPs: $F_{\text{total}} \leftarrow \sum_{l=1}^{L} F_l \times r_l$
5: Build a min-heap $H$ with key $S_l \times (1 - r_l)$ for all blocks
6: **while** $F_{\text{total}} > F_{\text{target}}$ **do**
7:      $l^* \leftarrow H.\text{pop}()$          // block with smallest $S_l \times (1 - r_l)$
8:      $r_{l^*} \leftarrow r_{l^*} - \Delta r$
9:      $F_{\text{total}} \leftarrow F_{\text{total}} - F_l \times \Delta r$
10:      $H.\text{push}(S_{l^*} \times (1 - r_{l^*}))$
11: **end while**
12: **return** $\{r_l\}_{l=1}^{L}$

---

# C. Allocated Retention Ratio across Blocks

Figure 4 shows the block-wise retention ratios allocated by the sensitivity-aware retention ratio allocation algorithm across different models, fine-tuning methods, and datasets. Across all settings, later blocks receive higher retention ratios than earlier ones as a direct result of cumulative token selection. In addition, the retention ratio of early blocks reaches zero in all configurations, indicating that backward computation for all tokens in those blocks is entirely skipped.

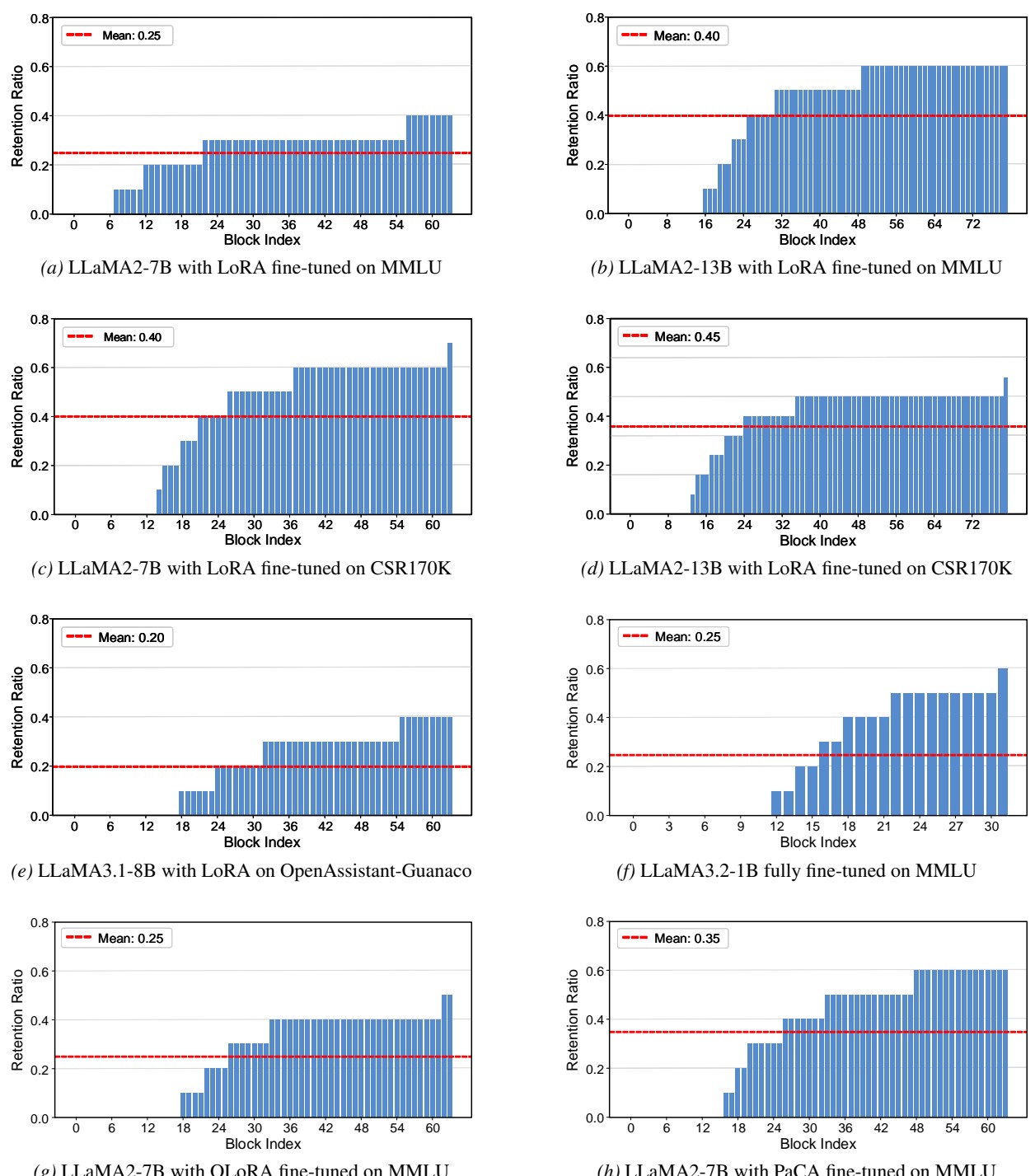

*(a)* LLaMA2-7B with LoRA fine-tuned on MMLU

*(b)* LLaMA2-13B with LoRA fine-tuned on MMLU

*(c)* LLaMA2-7B with LoRA fine-tuned on CSR170K

*(d)* LLaMA2-13B with LoRA fine-tuned on CSR170K

*(e)* LLaMA3.1-8B with LoRA on OpenAssistant-Guanaco

*(f)* LLaMA3.2-1B fully fine-tuned on MMLU

*(g)* LLaMA2-7B with QLoRA fine-tuned on MMLU

*(h)* LLaMA2-7B with PaCA fine-tuned on MMLU

*Figure 4.* Block-wise retention ratios for different LLaMA variants, fine-tuning methods (LoRA, QLoRA, PaCA, full fine-tuning), and datasets. The red dashed line denotes the mean retention ratio.

# D. Training Efficiency and Accuracy Trade-Off

## D.1. Results on Multitask Language Understanding

We analyze the trade-off between training efficiency and task performance on 5-shot MMLU for LLaMA2-7B. Table 10 and Figure 5 report results across various training configurations, revealing the full trade-off space between peak memory usage, training speedup, and accuracy. Across configurations, TokenDrop consistently achieves superior trade-offs, maintaining higher MMLU accuracy under more aggressive memory reduction and training acceleration compared to DropBP and TokenTune.

*Table 10.* Trade-off between training efficiency and 5-shot MMLU accuracy for LLaMA2-7B fine-tuning using different backpropagation skipping methods.

| Model | Method | Peak Memory | SpeedUp | Accuracy (%) | | | | |
|---|---|---|---|---|---|---|---|---|
| | | | | Hum. | STEM | Social | Other | Avg. |
| **LLaMA2-7B** | LoRA | 63.9 GB | 1.00× | 50.5 | 42.4 | 61.4 | 59.5 | 53.1 |
| | LoRA+DropBP | 54.0 GB | 1.13× | 50.3 | 42.8 | 62.2 | 59.9 | 53.3 |
| | LoRA+DropBP | 51.2 GB | 1.22× | 49.3 | 41.3 | 60.6 | 59.6 | 52.3 |
| | LoRA+DropBP | 49.9 GB | 1.29× | 48.0 | 41.3 | 58.4 | 58.1 | 51.0 |
| | LoRA+DropBP | 45.5 GB | 1.38× | 47.0 | 39.7 | 56.5 | 56.3 | 49.5 |
| | LoRA+TokenTune | 48.3 GB | 1.14× | 48.1 | 41.8 | 59.5 | 58.6 | 51.5 |
| | LoRA+TokenTune | 45.2 GB | 1.18× | 47.9 | 41.9 | 58.7 | 58.4 | 51.3 |
| | LoRA+TokenTune | 38.8 GB | 1.31× | 46.4 | 40.8 | 57.9 | 57.8 | 50.1 |
| | LoRA+TokenTune | 36.7 GB | 1.36× | 45.6 | 40.2 | 56.3 | 56.8 | 49.2 |
| | LoRA+TokenDrop | 48.8 GB | 1.16× | 50.7 | 43.4 | 62.5 | 60.4 | 53.8 |
| | LoRA+TokenDrop | 44.9 GB | 1.24× | 50.1 | 44.0 | 62.6 | 60.3 | 53.7 |
| | LoRA+TokenDrop | 38.1 GB | 1.35× | 51.2 | 43.7 | 62.7 | 60.3 | 54.1 |
| | LoRA+TokenDrop | 34.7 GB | 1.46× | 48.5 | 40.1 | 56.2 | 56.9 | 50.1 |

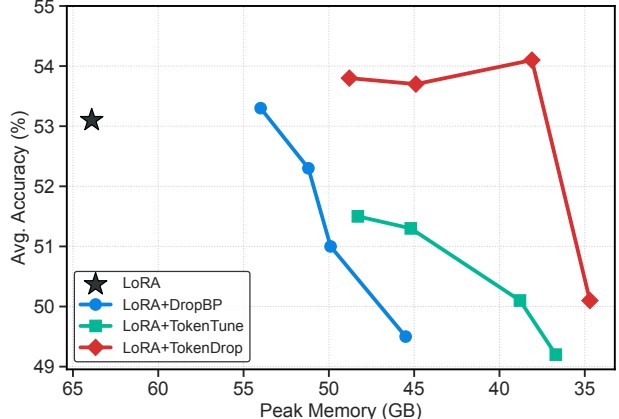

*(a)* Trade-off between peak memory and MMLU accuracy

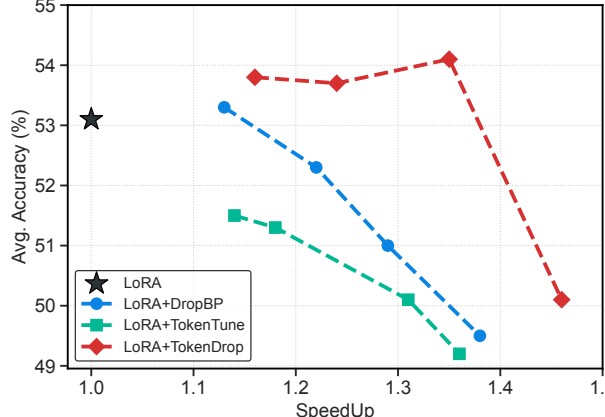

*(b)* Trade-off between training speedup and MMLU accuracy

*Figure 5.* 5-shot MMLU accuracy comparison of different backpropagation skipping methods for LLaMA2-7B under varying training efficiency settings.

## D.2. Results on Commonsense Reasoning

We further evaluate the training efficiency and accuracy trade-off on commonsense reasoning tasks using LLaMA2-7B. Table 11 reports detailed results across multiple configurations, while Figure 6 illustrates the relations between peak memory usage, training speedup, and 0-shot commonsense reasoning accuracy for different backpropagation skipping methods. TokenDrop consistently occupies the upper-right region of the plots in Figures 6a and 6b, achieving higher accuracy under both lower memory footprints and larger training speedups compared to DropBP and TokenTune.

*Table 11.* Trade-off between training efficiency and 0-shot commonsense reasoning accuracy for LLaMA2-7B fine-tuning using different backpropagation skipping methods.

| Model | Method | Peak Memory | SpeedUp | Accuracy (%) | | | | | | | | |
|-------|--------|-------------|---------|-------|-------|-------|-------|------|------|------|------|------|
| | | | | ARC-c | ARC-e | BoolQ | Hella | OBQA | PIQA | SIQA | Wino | Avg. |
| | LoRA | 44.9 GB | 1.00× | 55.2 | 82.5 | 79.6 | 80.0 | 51.3 | 81.8 | 59.0 | 83.1 | 71.6 |
| | LoRA+DropBP | 39.5 GB | 1.07× | 55.0 | 81.7 | 82.8 | 79.7 | 49.3 | 81.4 | 58.8 | 82.1 | 71.3 |
| | LoRA+DropBP | 36.5 GB | 1.14× | 54.3 | 81.3 | 82.6 | 79.4 | 47.5 | 80.9 | 58.0 | 80.4 | 70.5 |
| | LoRA+DropBP | 35.7 GB | 1.21× | 53.5 | 80.9 | 79.8 | 79.3 | 47.9 | 80.9 | 57.4 | 80.5 | 70.0 |
| | LoRA+DropBP | 33.4 GB | 1.29× | 52.4 | 80.5 | 78.3 | 78.7 | 46.3 | 80.5 | 57.4 | 78.7 | 69.1 |
| LLaMA2-7B | LoRA+TokenTune | 38.7 GB | 1.06× | 54.4 | 82.1 | 76.0 | 79.6 | 49.5 | 81.1 | 58.3 | 82.3 | 70.4 |
| | LoRA+TokenTune | 35.2 GB | 1.12× | 53.2 | 81.6 | 80.8 | 79.4 | 47.7 | 81.1 | 58.3 | 81.7 | 70.5 |
| | LoRA+TokenTune | 33.4 GB | 1.18× | 53.7 | 81.9 | 80.5 | 79.0 | 47.5 | 81.1 | 58.2 | 82.0 | 70.5 |
| | LoRA+TokenTune | 30.7 GB | 1.23× | 52.1 | 81.3 | 80.3 | 79.0 | 46.1 | 80.7 | 58.1 | 81.5 | 69.9 |
| | LoRA+TokenDrop | 38.1 GB | 1.09× | 54.2 | 82.0 | 81.5 | 79.6 | 50.9 | 81.4 | 58.5 | 82.4 | 71.3 |
| | LoRA+TokenDrop | 35.3 GB | 1.15× | 54.2 | 81.5 | 83.5 | 79.4 | 50.4 | 81.6 | 58.4 | 82.4 | 71.4 |
| | LoRA+TokenDrop | 32.9 GB | 1.23× | 54.0 | 81.5 | 85.2 | 79.0 | 49.2 | 81.0 | 58.1 | 82.2 | 71.3 |
| | LoRA+TokenDrop | 29.6 GB | 1.31× | 52.8 | 80.8 | 82.8 | 78.2 | 46.8 | 80.5 | 56.9 | 81.0 | 70.0 |

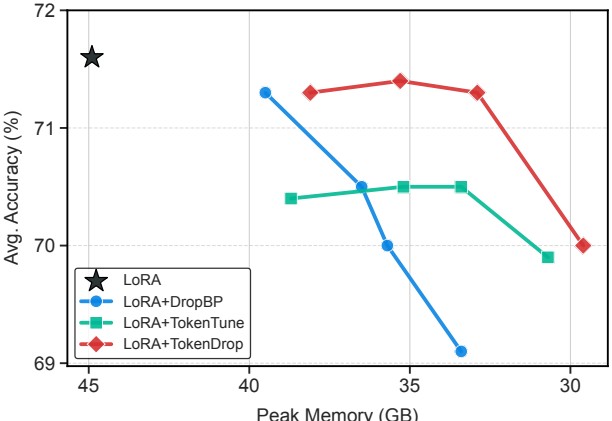

(a) Trade-off between peak memory and CSR170K accuracy

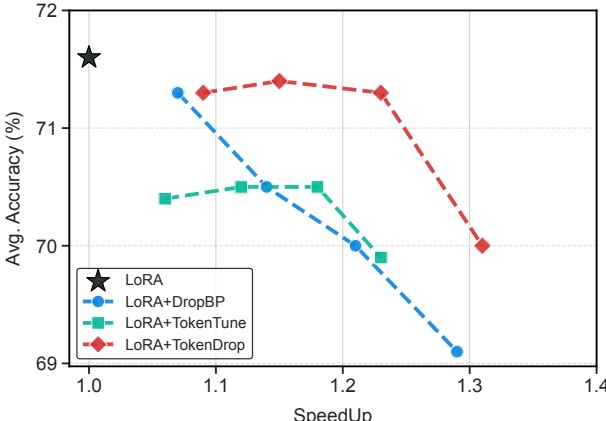

(b) Trade-off between training speedup and CSR170K accuracy

*Figure 6.* Accuracy comparison of different backpropagation skipping methods on commonsense reasoning tasks for LLaMA2-7B under varying training efficiency settings.

# E. Results on Long-Context Benchmark

To further evaluate the applicability of TokenDrop to long-context training, we conducted additional fine-tuning experiments on LongAlpaca (Chen et al., 2024) and evaluated the resulting models on LongBench (Bai et al., 2024). Specifically, we fine-tuned LLaMA3.2-1B with LoRA using a maximum sequence length of 16,384.

As shown in Table 12, TokenDrop lowers peak memory usage from 50.7 GB to 33.7 GB compared to LoRA, and provides a $1.23\times$ training speedup. At the same time, TokenDrop improves the average LongBench score from 22.6 to 23.5, matching the highest average score among the evaluated methods. We note that DropBP achieves a slightly higher wall-clock speedup in this setting. This is because, in long-context training, attention computation becomes increasingly dominant due to its quadratic dependence on sequence length, making the realized speedup more sensitive to attention-kernel efficiency. DropBP performs block-level backward skipping, whose efficiency is less dependent on attention-kernel implementation details, whereas token-level backward skipping in TokenDrop is not yet fully optimized in current FlashAttention implementations. Developing kernel-level support for token-level backward skipping is an important direction for future work and could further improve the practical efficiency of TokenDrop in long-context training.

*Table 12.* Comparison of peak memory usage, training speedup, and LongBench score when fine-tuning LLaMA3.2-1B on LongAlpaca.

| Method | Memory | SpeedUp | Multi-doc QA | Single-doc QA | Summ. | Few Shot | Synthetic | Code | Avg. |
|---|---|---|---|---|---|---|---|---|---|
| Not Tuned | - | - | 9.5 | 10.8 | 16.3 | 44.1 | 2.3 | 24.7 | 18.1 |
| LoRA | 50.7 GB | $1.00\times$ | $15.6_{\pm 0.6}$ | $24.5_{\pm 0.6}$ | $13.6_{\pm 4.1}$ | $54.7_{\pm 2.0}$ | $4.0_{\pm 0.3}$ | $17.2_{\pm 1.9}$ | $22.6_{\pm 1.4}$ |
| LoRA+DropBP | 41.2 GB | $\mathbf{1.29\times}$ | $18.7_{\pm 3.3}$ | $27.7_{\pm 1.6}$ | $14.9_{\pm 2.5}$ | $53.4_{\pm 1.6}$ | $4.2_{\pm 1.0}$ | $14.8_{\pm 5.7}$ | $\mathbf{23.5_{\pm 1.3}}$ |
| LoRA+TokenTune | 34.5 GB | $1.07\times$ | $12.5_{\pm 2.3}$ | $17.9_{\pm 1.8}$ | $12.9_{\pm 3.3}$ | $48.8_{\pm 3.2}$ | $2.6_{\pm 1.9}$ | $12.4_{\pm 4.2}$ | $18.8_{\pm 0.5}$ |
| LoRA+TokenDrop | **33.7 GB** | $1.23\times$ | $18.3_{\pm 2.2}$ | $25.6_{\pm 0.8}$ | $15.6_{\pm 3.8}$ | $54.8_{\pm 1.6}$ | $3.8_{\pm 0.3}$ | $16.1_{\pm 3.4}$ | $\mathbf{23.5_{\pm 1.5}}$ |

# F. Comparison to Activation Checkpointing

Activation checkpointing (AC) is a widely used technique for reducing activation memory during training. While both AC and TokenDrop reduce activation memory, they achieve this goal through different mechanisms. AC reduces memory usage by storing only a subset of activations during the forward pass and recomputing the discarded activations during backpropagation. As a result, AC introduces additional recomputation overhead. In contrast, TokenDrop reduces both activation memory and computational cost by skipping backward operations for less important tokens.

To make this distinction explicit, Table 13 summarizes the peak activation memory and FLOPs of AC and TokenDrop. Here, $L$ denotes the number of MHA and FFN blocks, $M_{\text{Input}}$ the activation size of a layer input, $M_{\text{MHA}}$ the activation size within an MHA block, $M_{\text{FFN}}$ the activation size within an FFN block, $F_l^{\text{bwd}}$ the backward FLOPs of block $l$, and $r_l \in [0, 1]$ the token retention ratio at block $l$ in TokenDrop. For each block $l$, $M_l$ corresponds to either $M_{\text{MHA}}$ or $M_{\text{FFN}}$, depending on the block type. We denote the total forward and backward FLOPs of the full model by $F_{\text{full}}^{\text{fwd}}$ and $F_{\text{full}}^{\text{bwd}}$, respectively. We consider a conventional AC implementation in LLM training frameworks, where checkpointing is applied at the Transformer layer level. In this setting, only layer inputs are stored, while intermediate activations inside each layer are discarded. During backpropagation, the forward computation must be recomputed to recover these intermediate activations, effectively doubling the forward computation cost.

*Table 13.* Comparison of peak activation memory and FLOPs between activation checkpointing and TokenDrop.

| Metric | Activation Checkpointing | TokenDrop |
|---|---|---|
| Peak Activation Memory | $\frac{L}{2} \cdot M_{\text{Input}} + (M_{\text{MHA}} + M_{\text{FFN}})$ | $\sum_{l=1}^{L-1} r_l \cdot M_l + M_{\text{FFN}}$ |
| FLOPs | $2 \cdot F_{\text{full}}^{\text{fwd}} + F_{\text{full}}^{\text{bwd}}$ | $F_{\text{full}}^{\text{fwd}} + \sum_{l=1}^{L} r_l \cdot F_l^{\text{bwd}}$ |

Although our current implementation is not directly combined with AC, the two approaches are conceptually complementary. If TokenDrop is combined with AC, the dropped tokens do not participate in backpropagation, and therefore their intermediate activations do not need to be recomputed. Consequently, activation recomputation under AC would be required only for retained tokens, potentially reducing the overall recomputation overhead.

## G. Hyperparameter Settings for Experiments

For all experiments using LoRA or QLoRA, we set the LoRA dropout to zero and applied adapters to all projection matrices, including Q, K, V, O, Up, Down, and Gate. For PaCA, an adapter-free PEFT method, we applied it to the same set of projection matrices.

*Table 14.* Hyperparameters for multitask language understanding, commonsense reasoning, and instruction following tasks.

| Task | Model | Method | PEFT | Max Seq. Len. | Rank | Alpha | LR | Batch Size | Micro Batch Size | Epochs | Optimizer | LR Scheduler | Warmup Ratio |
|---|---|---|---|---|---|---|---|---|---|---|---|---|---|
| Multitask Language Understanding | LLaMA2-7B | Baseline DropBP TokenTune TokenDrop | LoRA | 512 | 32 | 64 | {5e-5, 1e-4, 2e-4} | 16 | 16 | 1 | AdamW | Cosine | 0.04 |
| | | Baseline TokenDrop | QLoRA | 512 | 32 | 64 | {5e-5, 1e-4, 2e-4} | 16 | 16 | 1 | AdamW | Cosine | 0.04 |
| | | Baseline TokenDrop | PaCA | 512 | 32 | 64 | {5e-5, 1e-4, 2e-4} | 16 | 16 | 1 | AdamW | Cosine | 0.04 |
| | LLaMA2-13B | Baseline DropBP TokenTune TokenDrop | LoRA | 512 | 32 | 64 | {5e-5, 1e-4, 2e-4} | 8 | 8 | 1 | AdamW | Cosine | 0.04 |
| | LLaMA3.2-1B | Baseline TokenDrop | Full-FT | 512 | - | - | {5e-5, 1e-4, 2e-4} | 32 | 32 | 1 | AdamW | Cosine | 0.04 |
| Commonsense Reasoning | LLaMA2-7B | Baseline DropBP TokenTune TokenDrop | LoRA | 512 | 32 | 64 | {5e-5, 1e-4, 2e-4} | 64 | 64 | 1 | AdamW | Cosine | 0.04 |
| | LLaMA2-13B | Baseline DropBP TokenTune TokenDrop | LoRA | 512 | 32 | 64 | {5e-5, 1e-4, 2e-4} | 48 | 48 | 1 | AdamW | Cosine | 0.04 |
| Instruction Following | LLaMA3.1-8B | Baseline DropBP TokenTune TokenDrop | LoRA | 1024 | 32 | 64 | {5e-5, 1e-4, 2e-4} | 32 | 16 | 1 | AdamW | Cosine | 0.04 |
| Mathematical Reasoning | LLaMA3.1-8B | Baseline DropBP TokenTune TokenDrop | LoRA | 1024 | 32 | 64 | {5e-5, 1e-4, 2e-4} | 16 | 16 | 1 | AdamW | Cosine | 0.04 |
| Code Generation | LLaMA3.1-8B | Baseline DropBP TokenTune TokenDrop | LoRA | 1024 | 32 | 64 | {5e-5, 1e-4, 2e-4} | 16 | 16 | 1 | AdamW | Cosine | 0.04 |
| Long-context Multitask | LLaMA3.2-1B | Baseline DropBP TokenTune TokenDrop | LoRA | 16384 | 32 | 64 | {5e-5, 1e-4, 2e-4} | 32 | 1 | 1 | AdamW | Cosine | 0.04 |

