# OpenReview forum: "TokenDrop: Token-Level Importance-Aware Backward Propagation Skipping for Efficient LLM Fine-Tuning"
_ICML.cc/2026/Conference — ICML 2026 regular_

### Official Review · Reviewer_jZnc · 2026-02-23

**Soundness:** 3
**Presentation:** 3
**Significance:** 2
**Originality:** 2
**Overall Recommendation:** 3
**Confidence:** 3

**Summary:**

This paper introduces TokenDrop, a novel backpropagation-skipping technique designed to reduce the activation memory footprint and computational overhead of fine-tuning Large Language Models (LLMs). Unlike prior methods that bypass gradient computations at the coarse block level or randomly discard tokens, TokenDrop selectively skips the backward pass for less informative tokens based on their actual contribution to the learning process. To achieve this efficiently, the authors propose a lightweight, forward-only scoring metric that evaluates token importance using the l2-norm of their residual updates. To ensure training stability and optimal token evaluation, the framework incorporates cumulative token selection to maintain gradient continuity across layers, alongside lazy selection scheduling to make globally informed retention decisions only when memory budgets approach their limits. Extensive experiments across diverse tasks demonstrate that TokenDrop significantly accelerates training and reduces peak memory usage while maintaining or even improving downstream task accuracy, proving its broad compatibility with paradigms like full fine-tuning, LoRA, and QLoRA.

**Compliance With Llm Reviewing Policy:**

Affirmed.

**Final Justification:**

My final recommendation remains unchanged after considering the paper, the rebuttal, and the follow-up response. The additional experiments on Qwen2.5, Mistral, and LLaMA2-70B are helpful and improve the paper’s credibility, and I appreciate the authors’ effort to address my concerns about model diversity and scaling behavior. However, the new evidence is still somewhat limited in scope: the non-LLaMA validation remains narrow, the 70B result is based on a partial training setting rather than a full large-scale evaluation, and the explanation for the consistent performance gains over the fine-tuning baselines remains largely qualitative. Overall, while the responses strengthen the paper, they do not fully resolve my main concerns regarding significance, scaling, and generalizability, so my overall evaluation and score remain unchanged.

**Key Questions For Authors:**

1. Inconsistency in Model Selection Across Tasks: Could the authors clarify the rationale behind using different base models for different tasks? Specifically, LLaMA2-7B and LLaMA2-13B were used for Multitask Language Understanding (MMLU) and Commonsense Reasoning , whereas LLaMA3.1-8B was selected for the Instruction Following task.

2. Counterintuitive Performance Improvements: The baselines DropBP and TokenTune generally result in performance degradation compared to standard LoRA. However, TokenDrop not only matches the baseline but frequently improves upon it (e.g., LLaMA2-7B on MMLU improves from 53.1% to 54.1% ). Furthermore, this performance enhancement is observed across other paradigms like Full-FT (which jumps from 36.0% to 46.1%), QLoRA, and PaCA. Could the authors provide a deeper analysis of why TokenDrop acts as a performance enhancer? Does the l2-norm criterion inadvertently act as a regularizer that filters out noisy or detrimental tokens during training?

3. Variable Margins of Improvement Across Tasks: Observing Table 2 (Commonsense Reasoning), TokenDrop achieves an average accuracy of 71.3% for LLaMA2-7B, which is a relatively modest improvement over DropBP (70.0%) and TokenTune (70.5%) compared to the wider gaps seen in MMLU. Could the authors analyze why TokenDrop's relative advantage diminishes on commonsense reasoning benchmarks? Are there specific task characteristics (e.g., reasoning-heavy vs. knowledge retrieval-heavy) where the proposed importance scoring metric is fundamentally limited or less effective?

**Limitations:**

The authors have not adequately discussed the limitations of their work. I recommend that the authors add a brief discussion outlining specific scenarios—such as particular reasoning-heavy task types, distinct model architectures (e.g., MoE), or larger model scales—where the effectiveness of TokenDrop might be constrained or reduced, as this would greatly clarify the method's practical applicability boundaries.

**Strengths And Weaknesses:**

**Strengths**
1. Originality & Significance (Identification of a Critical Gap). The paper identifies a crucial limitation in existing backpropagation skipping methods (such as DropBP and TokenTune): they overlook the heterogeneous importance of individual tokens during fine-tuning. By recognizing that not all tokens contribute equally to the learning process, the authors propose a highly motivated, token-level importance-aware skipping method that effectively reduces activation memory usage and computational cost.
2. Originality & Soundness (Methodological Innovation). The proposed TokenDrop framework introduces several innovations. Relying on the l2-norm of residual updates serves as a computationally free, forward-only metric for token importance. Furthermore, the introduction of cumulative token selection to preserve gradient integrity across depths, and lazy selection scheduling to defer selection for globally informed scoring, are technically sound solutions to the inherent instability of local token dropping.
3. Soundness (Comprehensive Empirical Validation). The experimental setup is rigorous and extensive. The authors validate TokenDrop across a diverse set of tasks, including Multitask Language Understanding, Commonsense Reasoning, and Instruction Following. The method is thoroughly compared against relevant  baselines (LoRA, DropBP, TokenTune). Furthermore, the authors demonstrate the method's versatility by proving its compatibility across different training paradigms, including Full Fine-Tuning, QLoRA, and PaCA.
4. Significance (Strong Performance Trade-offs). TokenDrop achieves highly impressive practical results. The method not only matches (and occasionally exceeds) the task performance of the standard LoRA baseline, but it simultaneously achieves substantial reductions in peak memory usage (up to 42.9%) and accelerates training speeds (up to 1.50×). This provides immediate practical utility for resource-constrained LLM fine-tuning.
5. Soundness (Thorough Ablation Studies). The paper includes complete and well-designed ablation studies that effectively isolate the contributions of its core mechanisms. The systematic evaluation of cumulative token selection, lazy selection scheduling, and different token importance criteria clearly justifies the authors' architectural choices.

**Weaknesses**
1. Soundness & Significance (Limited Model Diversity). The experiments are exclusively conducted on the LLaMA model family (LLaMA2-7B/13B and LLaMA3.1-8B). While LLaMA is a strong representative model, evaluating solely on one model family fails to sufficiently demonstrate the method's scalability and generalizability to other distinct LLM architectures (e.g., MoE models, Qwen).
2. Soundness (Inconsistent Experimental Setup). There is an unexplained inconsistency regarding the choice of base models across different tasks. Specifically, the authors utilize LLaMA2-7B and LLaMA2-13B for Multitask Language Understanding and Commonsense Reasoning, but switch to LLaMA3.1-8B for the Instruction Following task. A clear justification for this discrepancy should be provided to ensure rigorous comparative analysis.
3. Significance (Scaling Behavior on Larger Models). Observing Table 1, the performance robustness and accuracy improvements of TokenDrop appear more pronounced on LLaMA2-7B than on LLaMA2-13B(). This raises a concern: does the performance benefit or stability of TokenDrop gradually diminish as the model parameter scale increases? Given that PEFT methods are valuable for much larger models, the paper would be significantly strengthened by including experiments on larger-scale models to validate its scaling trajectory.
4. Soundness (Lack of Statistical Significance Analysis). Although Appendix E states that each experiment was conducted three times with different seeds and the results were averaged(), the main text and tables lack standard deviations, error bars, or formal statistical significance testing.
5. Presentation (Inconsistent Tense Usage and Typos). In Section 4 (Experiments), the text frequently mixes tenses, jumping between the past tense (e.g., "LoRA was adopted") and the present tense. Additionally, there is a prominent typo in Figure 1(a) where the y-axis is labeled "Memroy Usage" instead of "Memory Usage".

---

> ### Author Rebuttal · Authors · 2026-03-31
>
> We thank the reviewer for taking the time to carefully review our paper and for providing valuable feedback that helps strengthen our work.
>
> ---
>
> **Q4.1.** Limited Model Diversity
>
> **A4.1** We appreciate the reviewer for raising this concern regarding model diversity.  We conducted additional experiments on a **non-LLaMA architecture, Qwen2.5.**  As Qwen models are already near **performance saturation on moderate-difficulty benchmarks**, we instead consider a **code generation task**, where evaluation is sensitive to small prediction errors. Specifically, we conducted experiments using Qwen2.5-7B, Evol-Instruction-66k for fine-tuning, and HumanEval+ for evaluation. The details of the results are provided in **A2.4** (response to Reviewer veRb).
>
> The results show that TokenDrop consistently achieves **training acceleration and memory reduction while maintaining comparable accuracy**, demonstrating that its effectiveness is **not tied to the LLaMA family.** Due to time constraints, we limited our additional experiments to a single Qwen model. Broader validation (e.g., across different scales and MoE architectures) will be included in the camera-ready version if the papaer is accepted.
>
> ---
>
> **Q4.2.** Inconsistency in Model Selection
>
> **A4.2.** We thank the reviewer for pointing out this inconsistency in model selection. Our choices follow **established practices in prior works, including DropBP and TokenTune.**
>
> For MMLU and CSR170k, we adopted LLaMA2-7B/13B, as these exact configurations are **consistently used in prior method**s. Moreover, stronger models such as LLaMA3.1-8B tend to exhibit **performance saturation** on these tasks, leaving limited headroom to meaningfully assess improvements when fine-tuning. In addition, the LLaMA3 family lacks an intermediate-scale model (e.g., 13B), making it difficult to study scalability under our limited GPU resources.
>
> For instruction following, we used LLaMA3.1-8B, as it is more representative of **current-generation instruction-tuned evaluation settings**. For example, prior work such as DropBP used LLaMA3-8B, while SparseLoRA [1] employed LLaMA3.1-8B. This reflects a broader trend of leveraging recent LLaMA3-class models for instruction-following benchmarks, and our choice is aligned with this trend.
>
> ---
>
> **Q4.3.** Scaling Behavior
>
> **A4.3.** We thank the reviewer for this insightful observation.  We agree that the relative improvements are more pronounced on LLaMA2-7B than on 13B. However, this does not indicate diminished effectiveness of TokenDrop at larger scales, as TokenDrop primarily targets **training efficiency while preserving model quality**. In our experiments, even on LLaMA2-13B, TokenDrop consistently maintains **comparable accuracy to the baseline while achieving substantial efficiency gains, indicating that its stability is preserved at larger scales**. To further validate its scaling behavior, we are currently extending our experiments to **LLaMA2-70B with QLoRA** and aim to include these results during the discussion period.
>
> ---
>
> **Q4.4.** Lack of Statistical Significance Analysis
>
> **A4.4**  We thank the reviewer for raising this concern. To address this, we reported standard deviations with the averaged accuracies, provided in **A2.3** (response to Reviewer veRB). The observed variance is comparable to the baseline, supporting the stability of our method. We will include standard deviations for all results in the final version.
>
> ---
>
> **Q4.5.** Inconsistent Tense and Typos
>
> **A4.5.** We thank the reviewer for pointing out these issues. We will revise Section 4 to ensure consistent tense usage and correct the typo in the final version.
>
> ---
>
> **Q4.6.** Counter-Intuitive Performance Improvements
>
> **A4.6.** We thank the reviewer for this helpful question. The improvement can be interpreted as a combination of **implicit regularization and filtering effect**, where TokenDrop suppresses low-impact or noisy gradient contributions from less informative tokens. This is consistent with prior work, such as RHO-1 [2] and Token Cleaning [3], which show that focusing on learning informative tokens can improve performance.
>
> ---
>
> **Q4.7.** Variable Margins of Improvement Across Tasks
>
> **A4.7.** We thank the reviewer for this insightful question. We note that CSR170k has significantly shorter inputs (avg. 145) compared to MMLU (avg. 393), which may reduce the effective token budget for dropping. In other words, shorter sequences may contain **fewer redundant or low-impact tokens**, limiting the opportunity for TokenDrop to filter them.
>
> ---
>
> **References**
>
> [1] Khaki, S., et al. "Sparselora: Accelerating llm fine-tuning with contextual sparsity." arXiv preprint, 2025.
>
> [2] Lin, Z., et al. “Not all tokens are what you need for pretraining.” Advances in Neural Information Processing Systems, 2024.
>
> [3] Pang, J., et al. “Token cleaning: Fine-grained data selection for llm supervised fine-tuning.” arXiv preprint, 2025.

---

> > ### Author Rebuttal · Reviewer_jZnc · 2026-04-04
> >
> > Thank you for the rebuttal. The response is helpful and addresses part of my concerns, but my overall assessment remains unchanged.
> >
> > For (1) model diversity, the added Qwen2.5 result is a useful addition, but it is still limited to one extra model and one task, so I do not think it is sufficient to establish broad generalization beyond the LLaMA family. For (3) scaling behavior, the rebuttal acknowledges smaller gains on larger models, but does not yet provide direct evidence on substantially larger-scale settings, so the scaling trend remains unclear. For (6) the performance gains over the underlying fine-tuning baselines, the regularization/filtering explanation is plausible, but remains qualitative and does not fully explain the consistent improvements across settings.
> >
> > Overall, these remaining concerns are central to the paper’s significance and would require more substantial evidence in the paper itself. Therefore, my overall judgment and score remain unchanged.

---

> > > ### Author Response · Authors · 2026-04-07
> > >
> > > Dear Reviewer jZnc,
> > >
> > > We thank the reviewer for the thoughtful feedback and for acknowledging the improvements in our rebuttal. We address the remaining concerns below.
> > >
> > > ---
> > >
> > > **(1) Model Diversity**
> > >
> > > We agree that broader coverage would further strengthen the claim. To further address the concern on generalization of TokenDrop beyond the LLaMA family, we have extended our experiments to additional model families and tasks. Specifically, we have included **Qwen2.5-7B and Mistral-7B on math reasoning tasks.**
> > > Across these settings, TokenDrop consistently maintains its advantage in the accuracy-efficiency trade-off, demonstrating that the proposed method is not tied to a specific architecture family or task type.
> > >
> > > ***Table R4.1*. Comparison of peak memory usage, training speedup, and GSM8K-CoT score**
> > > | **Model** | **Method** | **Memory (GB)** | **SpeedUp** | **GSM8K-CoT** |
> > > | --- | --- | --- | --- | --- |
> > > | Qwen2.5-7B | LoRA | 56.5 | 1.00$\times$ | **78.9** |
> > > |  | DropBP | 46.8 | 1.25$\times$ | 75.1 |
> > > |  | TokenTune | 37.3 | 1.29$\times$ | 77.1 |
> > > |  | TokenDrop | **36.5** | **1.38**$\times$ | 78.7 |
> > > | Mistral-7B | LoRA | 50.5 | 1.00$\times$ | 52.3 |
> > > |  | DropBP | 40.4 | 1.28$\times$ | 48.7 |
> > > |  | TokenTune | 29.8 | 1.32$\times$ | 45.5 |
> > > |  | TokenDrop | **29.1** | **1.39**$\times$ | **53.5** |
> > >
> > > ---
> > >
> > > **(2) Scaling Behavior**
> > >
> > > We have additionally evaluated **LLaMA2-70B with QLoRA fine-tuning on MMLU**.
> > > Due to the substantial computational cost, we have conducted experiments on a 12.5% subset of one epoch, which still provides a meaningful proxy for training dynamics. As shown in ***Table R4.2***, TokenDrop **outperforms prior methods in the accuracy–efficiency trade-off**, suggesting that the benefits persist at larger scales. We note that the observed speedup is relatively smaller in this setting, primarily due to the overhead of quantization/dequantization in QLoRA, which reduces the proportion of time dominated by the backward computation.
> > > We will extend this experiment to a **full 1-epoch training** and include this discussion on scalability in the final version if accepted.
> > >
> > > ***Table R4.2*. Comparison of peak memory usage, training speedup, and MMLU accuracy**
> > >
> > > | **Model** | **Method** | **Memory (GB)** | **SpeedUp** | **Hum.** | **STEM** | **Social** | **Other** | **Avg.** |
> > > | --- | --- | --- | --- | --- | --- | --- | --- | --- |
> > > | LLaMA2-70B | QLoRA | 56.1 | 1.00$\times$ | 66.3 | 58.1 | 79.9 | 74.4 | **69.2** |
> > > |  | DropBP | 51.6 | 1.13$\times$ | 64.6 | 56.8 | 80.0 | 73.8 | 68.2 |
> > > |  | TokenTune | 47.1 | 1.05$\times$ | 62.2 | 56.3 | 80.1 | 73.7 | 67.4 |
> > > |  | TokenDrop | **46.5** | **1.15$\times$** | 65.7 | 57.6 | 79.6 | 74.1 | 68.8 |
> > >
> > > ---
> > >
> > > **(3) Explanation of Performance Gains**
> > >
> > > We acknowledge that our analysis remains primarily qualitative. Since a quantitative characterization of training dynamics and their impact on performance is inherently challenging in large-scale deep learning, we rely on controlled empirical evidence to support our interpretation. Our ablation study in ***Table 6*** provides direct evidence supporting this interpretation. Selecting tokens with large $l$2-norm consistently outperforms random selection, while selecting tokens with small $l$2-norm leads to worse performance than random selection. Together, these results indicate that the $l$2-norm captures meaningful token importance rather than acting as a heuristic. This observation is consistent with the structural role of residual updates: tokens with smaller residual updates tend to induce smaller contributions to fine-tuning, and thus removing them effectively filters low-impact signals.
> > >
> > > ---
> > >
> > > We sincerely thank the reviewer again for the constructive feedback and helpful suggestions, which have helped us improve the clarity of our work.
> > >
> > > Best regards,
> > >
> > > Authors of the paper #30274

---

### Official Review · Reviewer_f5aU · 2026-03-10

**Soundness:** 3
**Presentation:** 3
**Significance:** 3
**Originality:** 3
**Overall Recommendation:** 5
**Confidence:** 4

**Summary:**

This paper proposes TokenDrop, an importance-aware backward-skipping method for efficient LLM fine-tuning. The method estimates token importance from residual updates during the forward pass and skips backward computation for less important tokens. Additionally, it adds cumulative selection and lazy scheduling to improve stability and selection quality. The paper is clearly written and well contextualized relative to prior gradient-skipping work. Empirical results against DropBP and TokenTune are strong, with reported gains of up to 42.9% memory reduction and 1.50× speedup while largely preserving task performance.

**Compliance With Llm Reviewing Policy:**

Affirmed.

**Final Justification:**

I am increasing my recommendation to 5 after considering the rebuttal, and I also upweight my assessment of soundness to Good. The paper presents a clear, well-motivated, and practically relevant approach to improving the memory/speed/accuracy tradeoff in LLM fine-tuning, with strong empirical results and good presentation.
My main initial concerns were about evaluation breadth, especially generative tasks, stronger long-context evidence, and the lack of comparison/discussion around activation checkpointing. The rebuttal addressed these points meaningfully by adding new generative and long-context results and clarifying the comparison to checkpointing. These additions increased my confidence in the technical soundness and practical value of the work. While the evaluation could still be broader, I believe the authors sufficiently addressed my main concerns, and the paper meets the bar for acceptance.

**Key Questions For Authors:**

see weaknesses

**Limitations:**

yes

**Strengths And Weaknesses:**

The paper aims to reduce the memory and compute footprint of LLM fine-tuning, especially the activation-related memory bottleneck that remains even with common PEFT methods like LoRA.

Strengths:
- The core idea is simple, principled, and well positioned against relevant prior work.
- The method seems to improve the memory/speed/accuracy tradeoff over relevant baselines (TokenTune and DropBP) as measured by likelihood-based evaluation (lm-evaluation-harness) as well as generative tasks (LLM-as-Judge setup).

Weaknesses:
- the benchmark suite is still somewhat narrow for the paper’s practical claims. Most evaluations are likelihood-style benchmarks while MT-Bench is the only clearly generative evaluation and is judged by GPT-4o-mini. I would want to see more generative tasks with verifiable outcomes, such as GSM8K with CoT evaluation and HumanEval/MBPP, to better test whether TokenDrop preserves chain-of-thought reasoning and generation quality beyond multiple-choice-style scoring. This is especially important since LoRA finetuning is now very commonly applied during RL tuning / on-policy distillation of LLMs.
-  To continue the above, the long-context motivation is stronger than the long-context evidence. Main reported settings are short-context tasks with maximum sequence lenghts between 512-1024 tokens. Since this method is especially motivated by activation memory / compute pressure at longer sequence lengths, the paper could be strengthened by adding longer context finetuning tasks.
- A final practical weakness is that the paper does not compare against activation checkpointing or discuss its tradeoffs in enough detail. This matters because checkpointing is one of the simplest and most common ways practitioners reduce activation memory, and a direct comparison would make the practical value of TokenDrop much easier to assess. It would also help to explain more explicitly how TokenDrop composes with checkpointing, and how the FLOPs/memory tradeoff differs between them.

The paper makes a clear and practically relevant contribution, is well presented, and shows convincing improvements over the most directly relevant prior work. My main concerns are about evaluation breadth. The paper would be stronger with checkpointing baselines, long-context benchmarks, and verifiable generative evaluation. Overall, I view it as a useful contribution with some limitations that reduce impact but do not outweigh its merits.

---

> ### Author Rebuttal · Authors · 2026-03-31
>
> We appreciate the reviewer’s thorough assessment of our work and the constructive suggestions. Our detailed responses to the raised questions are given below.
>
> ---
>
> **Q3.1.** Experiments on Additional Generative Tasks
>
> **A3.1.** We thank the reviewer for this valuable suggestion. Following the reviewer's recommendation, we additionally evaluated TokenDrop on **generative tasks**, focusing on **mathematical reasoning** and **code generation**.
>
> - For **mathematical reasoning**, we fine-tuned LLaMA3.1-8B on MetaMathQA-40K and evaluated on GSM8K with chain-of-thought (CoT) prompting using *lm-evaluation-harness*.
> - For **code generation**, we fine-tuned LLaMA3.1-8B and Qwen2.5-7B on Evol-instruction-66k and evaluated on HumanEval+, which contains more test cases than HumanEval, using *EvalPlus*.
>
>
> ***Table R3.1*. Comparison of memory usage, training speedup, and accuracy on GSM8K-CoT**
>
>
> | Model       | Method        | Memory (GB) | SpeedUp          | GSM8K-CoT |
> | ----------- | ------------- | ----------- | ---------------- | --------- |
> | LLaMA3.1-8B | Not Tuned     |             |                  | 53.37     |
> |             | LoRA          | 54.2        | 1.00$\times$     | 55.72     |
> |             | DropBP        | 44.7        | 1.21$\times$     | 49.28     |
> |             | TokenTune     | 40.4        | 1.19$\times$     | 50.57     |
> |             | TokenDrop | **38.2**    | **1.27$\times$** | **56.10** |
>
>
> Please refer to ***Table R2.2*** in the response to Reviewer veRb for code generation results.
> Across both tasks, TokenDrop consistently **matches or exceeds LoRA in accuracy while significantly improving training efficiency.** Additionally, it achieves **better accuracy than prior backward-skipping methods**, with comparable efficiency. These results suggest that TokenDrop **generalizes beyond likelihood-based evaluation to generation-heavy tasks** with strict correctness criteria.
>
> ---
>
> **Q3.2.** Experiments on Long Context Tasks
>
> **A3.2.** We thank the reviewer for this valuable feedback. Following the suggestion, we conducted additional experiments on **long-context dataset with a maximum sequence length of 16,384**, and observed that TokenDrop maintains stable performance under extended sequence lengths. Please refer to **A1.3** (the response for Reviewer B9cU) for details.
>
> ---
>
> **Q3.3** Comparison to Activation Checkpointing
>
> **A3.3** We thank the reviewer for raising concerns about the lack of comparison with activation checkpointing. While both activation checkpointing (AC) and TokenDrop reduce activation memory during training, **AC introduces additional recomputation overhead**, whereas **TokenDrop reduces computational cost by skipping backward operations for less important tokens.**
>
> To make this comparison explicit, we formalize their peak activation memory and FLOPs in ***Table R3.2***. Here, $L$ denotes the number of MHA/FFN blocks, $M_{\text{Input}}$ the activation size of a layer input, $M_l$ the activation size at block $l$, $F_l^{\text{bwd}}$ the backward FLOPs of block $l$, and $r_l \in [0,1)$ the retention ratio at block $l$ in TokenDrop. $F_{\text{full}}^{\text{fwd}}$ and $F_{\text{full}}^{\text{bwd}}$ denote the total forward and backward FLOPs of the full model.
>
> We assume AC implementation in a conventional LLM framework (e.g., Huggingface transformers), where activation checkpointing is applied at the Transformer layer level, storing only layer inputs while discarding intermediate activations. During backpropagation, the forward pass must be recomputed to recover these activations, effectively doubling the forward computation cost.
>
> ***Table R3.2* Comparison of Peak Activation Memory and FLOPs between AC and TokenDrop**
>
> | Metric | Activation Checkpointing | TokenDrop |
> | --- | --- | --- |
> | **Peak Activation Memory** | $\frac{L}{2} \cdot M_{\text{Input}} + (M_{\text{MHA}} + M_{\text{FFN}})$ | $ \sum_{l=1}^{L-1} r_l \cdot M_l + M_{\text{FFN}}$ |
> | **FLOPs** | $2 \cdot F_{\text{full}}^{\text{fwd}} + F_{\text{full}}^{\text{bwd}}$ | $F_{\text{full}}^{\text{fwd}} + \sum_{l=1}^{L} r_l \cdot F_l^{\text{bwd}}$ |
>
> While our current implementation is not directly compatible with AC, **the two approaches are conceptually complementary**. When combined with AC, since TokenDrop excludes dropped tokens from backpropagation, their intermediate activations do not need to be recomputed. As a result, forward recomputation in AC would only be performed on retained tokens, reducing the overall overhead. We will include this discussion in the final version if the paper is accepted.

---

> > ### Author Rebuttal · Reviewer_f5aU · 2026-04-03
> >
> > I thank the authors for their additional experiments. The new results on generative evaluations and long context enhance the work and address my main concerns.
> >
> > I also appreciate the comparison with activation checkpointing. For me this represents a technically solid contribution with good evaluation and I will therefore increase my score accordingly to a 5.

---

> > > ### Author Response · Authors · 2026-04-06
> > >
> > > Dear Reviewer f5aU,
> > >
> > > We appreciate the reviewer’s positive feedback and the recognition that our additional experiments have addressed the key concerns. The reviewer’s suggestions significantly improved the quality of our work. In response, we expanded our evaluation to cover generative tasks and long-context settings, and clarified the comparison with activation checkpointing. These revisions strengthen both the empirical evidence and the overall presentation of our method. We will continue refining these parts for inclusion in the final version if the paper is accepted.
> > >
> > > We also appreciate the reviewer’s indication that the score will be increased. If deemed appropriate, we would be grateful if this could be reflected in the final score.
> > >
> > >
> > > Best regards,
> > >
> > > Authors of the paper #30274

---

### Official Review · Reviewer_veRb · 2026-03-12

**Soundness:** 3
**Presentation:** 3
**Significance:** 3
**Originality:** 2
**Overall Recommendation:** 4
**Confidence:** 3

**Summary:**

The paper proposes TokenDrop, a method to improve the efficiency of LLM fine-tuning by reducing activation storage and backward computation. The method skips gradient propagation for less important tokens, where token importance is estimated during the forward pass using the magnitude of residual updates from transformer blocks. To maintain training stability, the approach introduces cumulative token selection across layers and lazy selection scheduling that delays token filtering to accumulate importance over multiple layers. Experiments on MMLU, commonsense reasoning benchmarks, and instruction-following tasks show that TokenDrop reduces memory usage and training time while maintaining comparable performance to LoRA and outperforming prior backward-skipping methods.

**Compliance With Llm Reviewing Policy:**

Affirmed.

**Final Justification:**

TokenDrop addresses a bottleneck in LLM fine-tuning (activation memory and backward computation costs) that PEFT methods leave. The two proposed mechanisms are well-motivated and clearly presented, with supporting ablations. The method is complementary to LoRA and has broad practical relevance.

The rebuttal addressed all main concerns satisfactorily, with (1) Robustness analysis (Table R2.1, Appendix D),  (2) extra code generation results (Table R2.2), and (3) LLaMA2-70B experiments (Table R2.3) which confirm the efficiency–accuracy trade-off holds at scale, though speedup is reduced under QLoRA due to quantization overhead, and the experiment covers only ~ 13% of one epoch.
The authors correctly reframe the importance metric as a principled forward-pass proxy rather than a tight equivalence, and commit to clarifying in the final version.

The main remaining limitations are the heuristic nature of the importance metric and the preliminary 70B evaluation, which are acknowledged and do not undermine the core findings.

I stay at 4 (Weak Accept), primarily because the 70B results are preliminary and the importance metric, while better justified, is still an upper-bound proxy without direct gradient alignment validation.

**Key Questions For Authors:**

1. How and why is the magnitude of the residual update correlated with a token's contribution to training? Can you provide additional analysis or toy examples to better validate the claim?
2. Do you find scenarios or conditions in which the TokenDrop fails? Additionally, when might the self-attention approximation degrade training? Also, have you tested TokenDrop on larger models or longer-context tasks where token interactions may be more critical? e.g., coding LLM
3. Can you show how sensitive the method is to the retention schedule and token budget? Can you provide the training curves with variance across runs?

**Limitations:**

No. I do not find a sufficient discussion of the limitations of the approach. In particular, it would be helpful to discuss potential degradation and the scenarios where token dropping may harm model learning.

**Strengths And Weaknesses:**

**Strengths**

*Soundness*

1. The proposed approach (TokenDrop) to skip gradients for a subset of tokens is technically plausible and a reasonable strategy to reduce backward cost.
2. I really like the thoughtful design choices for cumulative selection and delayed eviction, which are logically motivated and simple to understand.
3. The paper provides ablation studies that help understand the role of the proposed components

*Presentation*
1. The paper is clear, with motivation, method components, and experiments in a logical order
2. The method description is relatively easy to follow

*Significance and Originality*
1. The paper studies an important and interesting problem of resolving major bottlenecks in LLM fine-tuning on activation storage and backward computation. The proposed method is complementary to existing PEFT methods, showing more potential practical benefit.
2. The paper has a novel combination of ideas (skipping gradients) with thoughtful training-stability mechanisms.



**Weaknesses**

*Soundness*
1. While the authors claim the impact of the approximation by skipping gradients is negligible, the paper provides limited empirical analysis quantifying the error introduced by this approximation.
2. The importance metric is heuristic and not fully justified. Though I understand the provided intuition for the use of the magnitude of residual updates, this is the main component and importance metric needs to be well validated and more empirical validation for why the magnitude of residual updates should correlate with a token’s contribution to training. The ablation study (on LLaMA2-7B) only shows the improvement over random selection but requires correlation, and also conducted on a simple setting.
3. Given the simplicity and wide applicability of the idea, the robustness analysis is quite limited. We need more tests for the sensitivity to hyperparameters, or random seeds, retention ratios
4. The paper does not discuss when token dropping could harm training. e.g., long-context reasoning or rare token learning.
5. TokenDrop still requires a full forward pass over all tokens, which may limit efficiency gains in settings where forward computation dominates training cost

*Presentation*
1. The key design choices need more explanations for better understanding, e.g., the sensitivity-based retention allocation (Section 3.4) is briefly described but lacks sufficient detail on implementation choices
2. The approximation of the attention gradients needs to be emphasized more, as it is an important component of the method

*Significance and Originality*
1. Experiment is limited mainly to a small set of LLaMA models and moderate context lengths.
2. The core idea is close to existing methods, so the conceptual novelty is moderate. The importance metric is somewhat heuristic without deeper insight into training dynamics.

---

> ### Author Rebuttal · Authors · 2026-03-31
>
> We thank the reviewer for their thorough evaluation and constructive feedback, which helped improve the clarity and quality of our work.
>
> ---
>
> **Q2.1.** Approximation of Gradients in Self-Attention
>
> **A2.1.** We thank the reviewer for highlighting this point. Appendix A shows that skipping the key/value gradients of dropped tokens only affects the corresponding weight updates under cumulative token selection. Empirically, ***Table 7*** shows no measurable degradation, indicating that the approximation error is negligible in practice. While one might expect the approximation to become harmful when certain tokens are heavily attended (e.g., long-context settings), additional long-context experiments (please see **A1.2**, response to Reviewer B9cU) show stable performance even in such scenarios.
>
> ---
>
> **Q2.2.** Limited Analysis of Importance Metric
>
> **A2.2.** We agree that the justification of the importance metric was not sufficiently developed in the manuscript. We have provided additional analysis to support it. Please refer to **A1.1** (response to Reviewer B9cU) for details.
>
> ---
>
> **Q2.3.** Robustness Analysis
>
> **A2.3.** We thank the reviewer for raising this concern. First, we clarify that all main results in the manuscript are **averaged over three random seeds**, and we additionally report **standard deviations** in ***Table R2.1*** to assess statistical stability. The observed deviation is comparable to the baseline, indicating stable performance across runs.
>
> ***Table R2.1*. Results of fine-tuning on MMLU with standard deviations.**
>
> | Model | Method | Hum. | STEM | Social | Other | Avg. |
> | --- | --- | --- | --- | --- | --- | --- |
> | LLaMA2-13B | LoRA | 55.81±0.23 | 47.11±0.51 | 66.77±0.63 | 64.29±0.51 | **58.13±0.35** |
> |  | TokenDrop | 55.92±0.79 | 46.65±0.17 | 66.07±0.68 | 64.13±0.36 | 57.88±0.40 |
>
> Regarding sensitivity to the token budget, Appendix D presents the **efficiency–accuracy trade-off across different retention ratios**, showing that TokenDrop maintains **stable performance over a wide range of budgets** while achieving favorable trade-offs. The mean retention ratios used in TokenDrop in ***Tables 8*** and ***9*** are (50%, 40%, 25%, 20%) and (60%, 50%, 40%, 30%), respectively.
>
> ---
>
> **Q2.4.** Failure Cases and Experiments on Code Generation
>
> **A2.4.** We thank the reviewer for this important question. **TokenDrop may degrade performance under very aggressive token budgets**, where too many tokens are removed and important signals are lost. As shown in ***Figures 5*** and ***6***, performance remains strong under moderate budgets but drops when the budget becomes overly constrained.
>
> We also evaluated TokenDrop on **code generation tasks** with Evol-instruction-66k dataset and  HumanEval+ benchmark. As shown in ***Table R2.2***, TokenDrop achieves memory reduction and speedup while maintaining accuracy, **demonstrating its applicability beyond language understanding tasks**.
>
> ***Table R2.2*. HumanEval+ evaluation results after fine-tuning with a maximum sequence length of 1024.**
>
> | Model | Method | Memory (GB) | SpeedUp | HumanEval+ |
> | --- | --- | --- | --- | --- |
> | LLaMA3.1-8B | LoRA | 66.8 | 1.00$\times$ | **36.6** |
> |  | DropBP | 53.5 | **1.24$\times$** | 33.5 |
> |  | TokenTune | 50.9 | 1.16$\times$ | 36.0 |
> |  | TokenDrop | **47.5** | **1.24$\times$** | **36.6** |
> | Qwen2.5-7B | LoRA | 66.4 | 1.00$\times$ | **75.0** |
> |  | DropBP | 55.7 | **1.21$\times$** | 73.2 |
> |  | TokenTune | **49.5** | 1.15$\times$ | 70.7 |
> |  | TokenDrop | 50.2 | 1.20$\times$ | **75.0** |
>
> ---
>
> **Q2.5.** Demand for Full Forward Pass
>
> **A2.5.** We thank the reviewer for this insightful comment. While TokenDrop does not reduce forward computation, training cost is largely dominated by the backward pass (≈2/3 in standard training, ≈1/2 with PEFT such as LoRA). Since TokenDrop reduces gradient computation, it still **provides meaningful efficiency gains despite requiring a full forward pass.**
>
> ---
>
> **Q2.6.** Details of Retention Ratio Allocation
>
> **A2.6.** We thank the reviewer for pointing this out. Detailed implementation of the retention ratio allocation strategy is provided in Appendix B. The method estimates block-wise sensitivity based on gradient deviation under token dropping, and allocates retention ratios via a greedy procedure, as shown in Algorithm 2.
>
> ---
>
> **Q2.7.** Larger-Scale Models
>
> **A2.7.** We thank the reviewer for this suggestion. Due to limited time and GPU resources, we are currently extending our evaluation to LLaMA2-70B, and aim to include these results during the discussion period.
>
> ---
>
> **Q2.8.** Loss Curves across Token Budgets
>
> **A2.8.** To address the reviewer’s request, we provide **training loss curves across different token budgets (i.e., mean retention ratio) with standard deviations across seeds**, demonstrating stable convergence and low variance across budgets.
>
> - [Loss_curves.png](https://drive.google.com/file/d/1rET_NTcx2wb3VoR1Gu9vHKCBLygu_Pr1/view?usp=drive_link)

---

> > ### Author Rebuttal · Reviewer_veRb · 2026-04-03
> >
> > I thank the authors for a thorough and responsive rebuttal. Most concerns are resolved:
> >
> > * Q2.1, Q2.6: Attention gradient approximation and retention allocation details are adequately addressed.
> > * Q2.3: Standard deviations (Table R2.1) and retention ratio sensitivity (Appendix D) display robustness.
> > * Q2.4: Code generation results (Table R2.2) broadens the applicability claim.
> > * Q2.5: I agree that backward pass dominates training cost is technically sound.
> >
> > Two points remain partially open:
> >
> > * Q2.2 (Importance metric): The Cauchy-Schwarz argument (shown in A1.1 of the Reviewer B9cU) is a useful theoretical motivation (norm of r_{l,t} upper-bounds a token's loss contribution) and justifying its use as a forward-pass proxy. However, the bound is inherently loose: the actual impact depends on the alignment, not norm alone, so to my best understanding, gradients are roughly uniform across tokens. Table 6 supports the proxy but does not distinguish it from other alternatives.
> > Neverthelss, I think including this derivation in the final paper will strengthen the paper.
> > * Q2.7 (Larger models): LLaMA2-70B results are still awating.
> >
> > I maintain my positive score of 4 (Weak Accept).

---

> > > ### Author Response · Authors · 2026-04-07
> > >
> > > Dear Reviewer veRb,
> > >
> > > We sincerely appreciate the reviewer’s positive feedback and the recognition that our responses have addressed most of the concerns. The reviewer’s comments have encouraged us to clarify the theoretical motivation of our importance metric and to further validate our approach across broader settings. We address the remaining points below.
> > >
> > > ---
> > >
> > > **Q2.2.** Limited Analysis of Importance Metric
> > >
> > > We agree that the current theoretical argument provides a loose bound and does not fully capture gradient alignment. Our goal is not to establish a tight equivalence, but to use it as an **efficient forward-pass proxy** for token importance.
> > > Despite this limitation, the metric is empirically effective: norm-based top-k selection outperforms random selection, and TokenDrop frequently achieves **equal or better performance than the baseline** across tasks and models.
> > > We will clarify this positioning and its limitations in the final version, following the reviewer’s suggestion.
> > >
> > > ---
> > >
> > > **Q2.7.** Larger-Scale Models
> > >
> > > We have additionally evaluated **LLaMA2-70B with QLoRA fine-tuning on MMLU**.
> > > Due to the substantial computational cost, we have conducted experiments on a 12.5% subset of one epoch, which still provides a meaningful proxy for training dynamics. As shown in ***Table R2.3***, TokenDrop **outperforms prior methods in the accuracy–efficiency trade-off**, suggesting that the benefits persist at larger scales. We note that the observed speedup is relatively smaller in this setting, primarily due to the overhead of **quantization/dequantization in QLoRA**, which reduces the proportion of time dominated by the backward computation.
> > > We will extend this experiment to a **full 1-epoch training** and include more comprehensive results in the final version if accepted.
> > >
> > > ***Table R2.3*. Comparison of peak memory usage, training speedup, and MMLU accuracy**
> > >
> > > | **Model** | **Method** | **Memory (GB)** | **SpeedUp** | **Hum.** | **STEM** | **Social** | **Other** | **Avg.** |
> > > | --- | --- | --- | --- | --- | --- | --- | --- | --- |
> > > | LLaMA2-70B | QLoRA | 56.1 | 1.00$\times$ | 66.3 | 58.1 | 79.9 | 74.4 | **69.2** |
> > > |  | DropBP | 51.6 | 1.13$\times$ | 64.6 | 56.8 | 80.0 | 73.8 | 68.2 |
> > > |  | TokenTune | 47.1 | 1.05$\times$ | 62.2 | 56.3 | 80.1 | 73.7 | 67.4 |
> > > |  | TokenDrop | **46.5** | **1.15$\times$** | 65.7 | 57.6 | 79.6 | 74.1 | 68.8 |
> > >
> > > ---
> > >
> > > Best regards,
> > >
> > > Authors of the paper #30274

---

### Official Review · Reviewer_B9cU · 2026-03-13

**Soundness:** 3
**Presentation:** 2
**Significance:** 3
**Originality:** 3
**Overall Recommendation:** 4
**Confidence:** 3

**Summary:**

The paper proposes **TokenDrop**, a token-level importance-aware backpropagation skipping method to reduce activation memory and accelerate LLM fine-tuning. The key contributions are:
1.A lightweight, gradient-free l2-norm metric that identifies uninformative tokens during the forward pass based on residual update magnitudes.
2.Cumulative selection and lazy scheduling strategies that ensure gradient continuity across layers and defer eviction for globally informed token retention.

**Compliance With Llm Reviewing Policy:**

Affirmed.

**Final Justification:**

The rebuttal addressed my main concerns, and so I kept my positive score.

**Key Questions For Authors:**

Please refer to the weaknesses outlined in the *Strengths and Weaknesses* above.

**Limitations:**

yes

**Strengths And Weaknesses:**

**Soundness**

* **Strength:**
The paper clearly acknowledges the approximation in self-attention (dropping K/V gradients for skipped tokens) and supports it with analysis and experiments (Appendix A, Table 7).

* **Weakness (Theoretical Depth):**
The proposed ℓ2-norm importance metric remains heuristic. Despite empirical validation, the paper lacks theoretical analysis explaining its correlation with gradient importance or training dynamics.

**Presentation:**

* **Strength:**
The paper is clearly written and well structured, with comprehensive experimental evaluation across multiple fine-tuning settings and diverse benchmarks. The ablation studies are thorough and the results consistently support the main claims.

* **Weakness(minor typo):**
“Memroy Usage [GB]” → “Memory”,

**Significance:**

* **Strength:**
The method is architecturally orthogonal to PEFT approaches such as LoRA, QLoRA, and PaCA, yielding consistent memory and speed gains without accuracy loss and remaining compatible with fused attention implementations such as FlashAttention-2.

* **Weakness(context limitation):**
The experiments are conducted with a maximum sequence length of 1024, and it is unclear whether the method generalizes to longer-context fine-tuning settings.


**Originality**

* **Strength:**
The work improves earlier backward-skipping methods by using importance-based token selection together with retention and scheduling, resulting in more stable behavior than block-level or random token dropping.


* **Weakness:**
The idea of token-level backward skipping has been explored in prior work. The main contribution here is the refinement of the selection and scheduling strategy.

---

> ### Author Rebuttal · Authors · 2026-03-31
>
> We thank the reviewer for carefully reviewing our submission and providing valuable feedback. Please see below for our response to the comments.
>
> ---
>
> **Q1.1.** Theoretical Justification of Importance Score
>
> **A1.1.**  We thank the reviewer for this important comment. In our setting, token importance **must be estimated during the forward pass, where gradient information is not available**. In this sense, the change in loss induced by a token provides a natural notion of importance. However, directly computing such loss variation would require backward information, making it infeasible in our setting.
>
> To approximate this, we leverage the residual structure of Transformer blocks. For a token $t$ at layer $l$, let the residual update be $r_{l,t} = f_l(x_{l,t})$, so that $x_{l+1,t} = x_{l,t} + r_{l,t}$. Here, we consider a local approximation where the effect of other tokens is fixed. The resulting loss change can be written as
> $\Delta L_{l,t} = L(x_{l,t} + r_{l,t}) - L(x_{l,t}).$
> By first-order Taylor expansion,
> $\Delta L_{l,t} \approx \nabla L(x_{l,t})^\top r_{l,t}. $
> Applying the Cauchy–Schwarz inequality gives
> $ |\Delta L_{l,t}| \le ||\nabla L(x_{l,t})||_2  ||r _ {l,t}||_2. $
>
> This shows that while the loss variation depends on both the gradient and the residual update, the **magnitude of the residual update directly controls its upper bound**. Since the gradient term is not accessible, we use $||r_{l,t}||_2$ as a practical proxy.
>
> It is not intended to directly measure the gradient importance of a token during training, but rather to serve as an efficient forward-pass proxy. Its effectiveness is supported by our empirical results, as shown in ***Table 6***: norm-based selection consistently outperforms random selection, while selecting small-norm tokens underperforms random, indicating that large-norm tokens are more informative.
>
> ---
>
> **Q1.2.** Typo in Figure 1(a)
>
> **A1.2.** We apologize for the oversight. The typo in Figure 1(a) will be corrected in the camera-ready version if the paper is accepted.
>
> ---
>
> **Q1.3.** Experiments on Long Context Scenarios
>
> **A1.3.** We thank the reviewer for raising this important concern. To address the concern, we conducted additional fine-tuning experiments using a long-context dataset. Specifically, we fine-tuned Llama 3.2-1B on LongAlpaca with **a maximum sequence length of 16,384**, then evaluated on LongBench. We intentionally used a relatively small model to minimize the memory footprint of model weights, thereby enabling training with significantly extended context lengths.
>
> ***Table R1.1*. Comparison of peak memory usage, training speedup, and LongBench score for Llama 3.2-1B.**
>
> | Method | Memory (GB) | SpeedUp | Multi-doc QA | Single-doc QA | Summarization | Few Shot | Synthetic | Code | Avg. |
> | --- | --- | --- | --- | --- | --- | --- | --- | --- | --- |
> | Not tuned |  |  | 9.46 | 10.81 | 16.26 | 44.12 | 2.31 | **24.73** | 18.05 |
> | LoRA | 50.7 | 1.00$\times$ | 15.83 | 24.11 | 18.28 | **56.98** | 4.33 | 19.38 | 24.13 |
> | DropBP | 41.2 | **1.29$\times$** | 18.10 | **29.49** | 17.70 | 55.15 | **5.22** | 16.97 | 24.96 |
> | TokenTune | 34.5 | 1.07$\times$ | 13.13 | 13.58 | 17.64 | 46.47 | 1.40 | 8.21 | 17.94 |
> | TokenDrop | **33.7** | 1.23$\times$ | **20.77** | 24.73 | **19.94** | 56.55 | 4.09 | 17.2 | **25.11** |
>
> ***Table R1.1*** shows that TokenDrop **reduces memory consumption by 33.5%** compared to LoRA and achieves a **1.23$\times$ speedup**, while attaining the **best average performance** on LongBench. Although DropBP shows slightly higher wall-clock speedup in this setting, we believe this is partly due to the current implementation limitation: token-level backward skipping is not fully optimized in existing FlashAttention kernels, whereas block-level skipping in DropBP is easier to realize efficiently. If kernel-level support for token-level backward skipping becomes available, the efficiency advantage of TokenDrop would likely become larger.
>
> Overall, these results demonstrate that TokenDrop generalizes effectively to long-context fine-tuning scenarios. We will include this discussion in the camera-ready version if the paper is accepted.
>
> ---
>
> **Q1.4.**  Modest Novelty
>
> **A1.4.** We agree that token-level backward skipping itself has been explored in TokenTune, and our method can be viewed as building upon this general direction. However, TokenTune relies on random token selection and does not address training stability.
> In contrast, we introduce (1) an effective importance metric derived from the residual structure of Transformers, and (2) a structured selection/scheduling framework that explicitly preserves gradient consistency and enables globally informed token selection.
> Empirically, these design choices lead to consistent improvements in both efficiency and performance across diverse benchmarks, indicating that our contribution is not merely incremental but provides a more reliable and effective framework for token-level backward skipping.

---

> > ### Author Rebuttal · Reviewer_B9cU · 2026-04-03
> >
> > Thanks for your detailed rebuttal, i keep my positive score.

---

> > > ### Author Response · Authors · 2026-04-06
> > >
> > > Dear Reviewer B9cU,
> > >
> > > We sincerely appreciate the reviewer's positive feedback and recognition that our responses have adequately addressed the reviewer's concerns. The reviewer's comments played an important role in strengthening our paper. In particular, they motivated us to refine the theoretical explanation of our token importance metric and extend our experimental section with long-context fine-tuning results. These additions provide stronger evidence for the effectiveness and generalizability of our approach. We will further refine and organize these discussions for inclusion in the final version of the paper if accepted.
> > >
> > > Best regards,
> > >
> > > Authors of the paper #30274

---

### Decision · Program_Chairs · 2026-04-30

**Decision:**

Accept (regular)

**Comment:**

This paper proposes TokenDrop, a token-level backward-skipping method for efficient LLM fine-tuning, using a lightweight forward-pass importance score together with cumulative selection and lazy scheduling. Reviewers agreed that the paper studies a relevant problem, is clearly written, and shows promising memory/speed improvements with limited loss in performance, and the rebuttal addressed several concerns by adding longer-context, generative/code, robustness, and preliminary larger-scale results. The main remaining concerns were the limited novelty relative to prior token-level skipping methods, the still heuristic and only loosely justified importance score, and the somewhat limited evidence on broader scaling and generalization. The AC also notes that the method still requires a full forward pass, so gains may be smaller in regimes where forward or implementation overhead dominates, and that stronger practical comparison against activation checkpointing would improve the paper. Overall, the idea is simple and reasonably effective, but some claims could be framed more conservatively in the final version.